# Novel Upper Bounds for the Constrained Most Probable Explanation Task

**Tahrima Rahman**       **Sara Rouhani**       **Vibhav Gogate**
The University of Texas at Dallas
{tahrima.rahman, sara.rouhani, vibhav.gogate@utdallas.edu}

## Abstract

We propose several schemes for upper bounding the optimal value of the constrained most probable explanation (CMPE) problem. Given a set of discrete random variables, two probabilistic graphical models defined over them and a real number $q$, this problem involves finding an assignment of values to all the variables such that the probability of the assignment is maximized according to the first model and is bounded by $q$ w.r.t. the second model. In prior work, it was shown that CMPE is a unifying problem with several applications and special cases including the nearest assignment problem, the decision preserving most probable explanation task and robust estimation. It was also shown that CMPE is NP-hard even on tractable models such as bounded treewidth networks and is hard for integer linear programming methods because it includes a dense global constraint. The main idea in our approach is to simplify the problem via Lagrange relaxation and decomposition to yield either a knapsack problem or the unconstrained most probable explanation (MPE) problem, and then solving the two problems, respectively using specialized knapsack algorithms and mini-buckets based upper bounding schemes. We evaluate our proposed scheme along several dimensions including quality of the bounds and computation time required on various benchmark graphical models and how it can be used to find heuristic, near-optimal feasible solutions in an example application pertaining to robust estimation and adversarial attacks on classifiers.

## 1   Introduction

We develop upper bounding algorithms for the *constrained most probable explanation* (CMPE) task [32], a recently defined unifying (discrete) optimization task over probabilistic graphical models (PGMs) [9, 20, 25] or log-linear models. At a high level, the CMPE task adds a *global capacity constraint* to the classic optimization problem in PGMs called *most probable explanation* (MPE). Given a set of log-potentials $\mathcal{F}$, namely a log-linear model, the MPE task seeks to find an assignment of values to all variables (i.e., an explanation) such that the sum over the projection of the assignment on the log-potentials is maximized, namely it has the maximum probability. The CMPE task adds a global constraint to MPE using a possibly different set of log-potentials $\mathcal{G}$ and a real number $q$. Specifically, it seeks to find the most probable assignment w.r.t. $\mathcal{F}$ under the constraint that the sum over the projection of the assignment on the log-potentials in $\mathcal{G}$ is smaller than or equal to $q$.

Our interest in CMPE stems from the fact that several tasks in Explainable AI [16] can be reduced to CMPE. For example, the nearest assignment problem [31] and the decision preserving explanation problem [7, 32] are instances of CMPE; the former seeks to find an assignment whose probability is as close as possible to a given assignment, namely, its nearest neighbor while the latter seeks to find the most probable extension of a given partial assignment according to a generative model such that the same (classification) decision is made according to a discriminative model. Other applications of CMPE include various queries in robust estimation [10] and detecting adversarial attacks.

35th Conference on Neural Information Processing Systems (NeurIPS 2021).

In terms of computational complexity, Rouhani et al. [31, 32] showed that CMPE is much harder than MPE. In particular, via a reduction from the multi-choice knapsack problem (MCKP) [19], Rouhani et al. showed that while MPE is linear time on graphical models having empty primal graphs, CMPE is NP-hard. However, despite these worst-case results, the good news is that MCKP is an instance of *easy NP-hard problems* [19] and can be well-approximated using advanced methods from the knapsack problems literature. The MCKP reduction also yields a graph-based technique for approximating CMPE. The key idea is to condition on variables, namely remove them from the primal graph $G$ until no more than $k$ nodes remain in each connected component. The removed nodes form a $k$-separator [3] and each assignment to them yields an MCKP. When the $k$-separator is bounded, this method yields a *fully polynomial time approximation scheme* (FPTAS). When the $k$-separator is not bounded, Rouhani et al. propose to perform local search yielding an anytime algorithm.

While the algorithms described above yield a lower bound, no non-trivial upper bounding algorithms are available for CMPE. Such algorithms serve two important purposes [24]: (1) generating heuristic near-optimal solutions; and (2) pruning the search space of branch and bound methods and thus improving their efficiency. In principle, CMPE can be encoded as a mixed integer linear program (MILP) and solved using MILP solvers such as Gurobi [17] and SCIP [1, 2]. However, CMPE and other related tasks such as MPE and MCKP are particularly difficult for MILP solvers because the global constraint in CMPE simultaneously restricts all the variables while MILP solvers are adept at handling sparse constraints. This motivates the development of specialized methods for CMPE.

In this paper, we propose two approaches for efficiently computing qualitative upper bounds for the CMPE task. These approaches relax either the objective or the constraint or both. Our first approach is based on *Lagrangian relaxation* (cf. [35]) and relaxes the global constraint using a Lagrange multiplier $\lambda \geq 0$ to yield an (unconstrained) MPE task. Given a value for $\lambda$, an upper bound on the MPE task yields an upper bound on CMPE. We propose to solve the MPE task using either exact or upper bounding approaches described in literature such as mini-bucket elimination [11], dual decomposition [14] and join graph based cost shifting schemes [18] and further tighten the upper bound by searching for the best possible value for $\lambda$, namely a value that minimizes the upper bound on MPE. Our second approach is based on *Lagrangian decomposition*. The key idea is to decompose the problem by duplicating variables [6] via equality constraints and then relaxing the latter using Lagrange multipliers such that the resulting problem reduces to MCKP. Solving the MCKP for a given assignment of values to the multipliers yields an upper bound on CMPE. We propose to further improve the bound by searching for the best possible value assignment to the multipliers via an iterative algorithm.

Empirically, we investigate the quality and computational efficiency of our proposed bounding techniques on a variety of CMPE tasks formulated on cutset networks [30] and graphical models used in the UAI competitions [13, 15]. We also explore a novel application of CMPE: making classifiers (expressed as log-linear models) change their decision by minimally changing the test example. We found that when there is a relatively small limit on the amount of storage space an algorithm can use, our approach based on Lagrange decomposition via MCKP yields the best upper bounds. However, it also requires significantly longer to converge. Conversely, when the space limit is relatively large, the approach based on Lagrangian relaxation that utilizes MPE solvers is superior.

## 2 Notation and Background

We denote *binary* discrete random variables using upper case letters (e.g., $X$, $Y$, etc), their sets using bold uppercase letters (e.g., $\boldsymbol{X}$, $\boldsymbol{Y}$, etc.) and assignment of values to them using bold lowercase letters (e.g., $\boldsymbol{x}$, $\boldsymbol{y}$, etc). If $\boldsymbol{S} \subseteq \boldsymbol{X}$, then $\boldsymbol{x_S}$ is the projection of $\boldsymbol{x}$ onto $\boldsymbol{S}$. We use letters $f$, $g$ and $h$ to denote log-potentials or features and the calligraphic capital letters such as $\mathcal{F}$, $\mathcal{G}$, etc. to denote sets of log-potentials. A log-linear model or a Markov network, denoted by $\mathcal{M}$ is a pair $\langle \boldsymbol{X}, \mathcal{F} \rangle$ where $\boldsymbol{X}$ is a set of variables and $\mathcal{F}$ is a set of log-potentials or features such that each feature $f_i \in \mathcal{F}$ is defined over a subset of variables $S(f_i) \subseteq \boldsymbol{X}$. $S(f_i)$ is called the *scope* of $f_i$. A log-potential maps each assignment in its scope to a real number ($\mathbb{R}$). A *primal graph* of $\mathcal{M}$ is an undirected graph $G(V, E)$ where $V$ and $E$ is the set of vertices and edges respectively such that each vertex $V_i \in V$ represents the random variable $X_i \in \boldsymbol{X}$ and there is an edge between vertices $V_i$ and $V_j$ in $G$ iff the corresponding variables appear together in the scope of some function in $\mathcal{F}$. Given an assignment $\boldsymbol{x}$, the weight of $\boldsymbol{x}$ w.r.t. $\mathcal{M}$, denoted by $\omega_{\mathcal{M}}(\boldsymbol{x})$ equals $\sum_{f \in \mathcal{F}} f(\boldsymbol{x}_{S(f)})$. For brevity, we will abuse notation and use $f(\boldsymbol{x})$ instead of $f(\boldsymbol{x}_{S(f)})$.

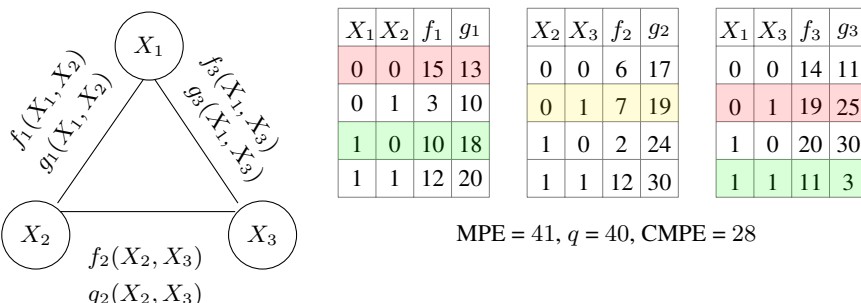

Figure 1: Combined primal graph of two log-linear models having log-potentials $\{f_1, f_2, f_3\}$ and $\{g_1, g_2, g_3\}$ respectively. Let $q = 40$, then the optimal solution $\boldsymbol{x}^*$ to the CMPE problem is $(X_1 = 1, X_2 = 0, X_3 = 1)$ (highlighted in green and yellow) and the optimal weight is 28. The MPE assignment having weight 41 is $(X_1 = 0, X_2 = 0, X_3 = 1)$ ( highlighted in red and yellow).

**Definition 1** (CMPE ). *Given two log-linear models $\mathcal{M}_1 = \langle \boldsymbol{X}, \mathcal{F} \rangle$ and $\mathcal{M}_2 = \langle \boldsymbol{X}, \mathcal{G} \rangle$ defined over the same set of variables $\boldsymbol{X}$ and a real value $q$, the constrained most-probable explanation (CMPE) task is to find an assignment $\boldsymbol{x}^*$ such that $w_{\mathcal{M}_1}(\boldsymbol{x}^*)$ is maximized and $w_{\mathcal{M}_2}(\boldsymbol{x}^*) \leq q$. Mathematically, the problem of computing the optimal weight is given by*

$$\max_{\boldsymbol{x}} \sum_{f \in \mathcal{F}} f(\boldsymbol{x}) \text{ s.t. } \sum_{g \in \mathcal{G}} g(\boldsymbol{x}) \leq q \tag{1}$$

$\boldsymbol{x}^*$ is the *optimal solution* to the CMPE problem. Let $c^* = \omega_{\mathcal{M}_1}(\boldsymbol{x}^*)$ denote the *optimal value* or weight. A *feasible solution* of CMPE is an assignment $\boldsymbol{x}$ s.t. $\sum_{g \in \mathcal{G}} g(\boldsymbol{x}) \leq q$.

The *combined primal graph* of CMPE is the graph obtained by taking the union of the edges of the primal graphs associated with $\mathcal{M}_1$ and $\mathcal{M}_2$ respectively. Formally, let $G_1(V, E_1)$ and $G_2(V, E_2)$ be the primal graphs associated with $\mathcal{M}_1$ and $\mathcal{M}_2$ respectively. Then the combined primal graph is $G(V, E_1 \cup E_2)$. Without loss of generality and for simplicity of exposition *we assume that both $\mathcal{M}_1$ and $\mathcal{M}_2$ have the same number of functions and for each function $f \in \mathcal{F}$ there is a corresponding function $g \in \mathcal{G}$ such that $S(f) = S(g)$ and vice versa*. Thus, under this assumption, $G_1 = G_2 = G$.

**Example 1.** *Fig. 1 shows two log-linear models that define a CMPE problem given $q = 40$.*

### 2.1 Prior Work on Upper Bounding the Most Probable Explanation Task

Removing the constraint in Eq. (1) encoded by $\mathcal{M}_2$ and $q$ yields the most probable explanation (MPE) task. MPE is known to be NP-hard in general but can be solved efficiently in practice when the primal graph has low treewidth or when the partition-based [11] upper bounding schemes yield close to optimal estimates within an AND/OR branch and bound framework [24]. In this paper and specifically in our experiments, we use mini-buckets based upper bounding schemes presented in Ihler et al. [18]. These schemes operate by first converting the primal graph of $\mathcal{M}$ to a join graph where each node called a cluster and each edge called a separator is associated and labeled with a subset of variables from $\mathcal{M}$. The join graph can be thought of as a relaxation of inference architectures called tree-decompositions (cf. [20]), on which MPE inference is exact, in that it satisfies all properties that the latter satisfies including the *running intersection property* except that it is a graph of clusters instead of a tree of clusters. Then these algorithms perform message-

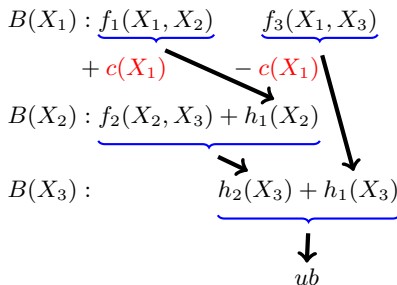

Figure 2: Message-passing in MB and MM on the log-linear model given in Fig. 1, having log-potentials $f_1$, $f_2$ and $f_3$.

passing on the edges of the join graph where the computational complexity of message-passing is exponential in the maximum cluster size. To control the complexity and yield a bounded polynomial time approximation, these schemes use an integer parameter $i$ known as the $i$-bound which bounds the number of variables in each cluster in the join graph by $i$.

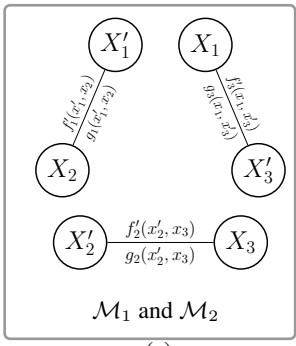

| $X_1'$ | $X_2$ | $f_1'$ | $g_1$ |
|---|---|---|---|
| 0 | 0 | 15 | 13 |
| 0 | 1 | 2 | 10 |
| 1 | 0 | 15.01 | 18 |
| 1 | 1 | 16.01 | 20 |

| $X_2'$ | $X_3$ | $f_2'$ | $g_2$ |
|---|---|---|---|
| 0 | 0 | 6 | 17 |
| 0 | 1 | 8 | 19 |
| 1 | 0 | 3 | 24 |
| 1 | 1 | 14 | 30 |

| $X_1$ | $X_3'$ | $f_3'$ | $g_3$ |
|---|---|---|---|
| 0 | 0 | 14 | 11 |
| 0 | 1 | 18 | 25 |
| 1 | 0 | 14.99 | 30 |
| 1 | 1 | 4.99 | 3 |

$q = 40$

| $i$ | $p_{1j}$ | $c_{1j}$ |
|---|---|---|
| $x_{11}$ | 15 | 13 |
| $x_{12}$ | 2 | 10 |
| $x_{13}$ | 15.01 | 18 |
| $x_{14}$ | 16.01 | 20 |

| $i$ | $p_{2j}$ | $c_{2j}$ |
|---|---|---|
| $x_{21}$ | 6 | 17 |
| $x_{22}$ | 8 | 19 |
| $x_{23}$ | 3 | 24 |
| $x_{24}$ | 14 | 30 |

| $i$ | $p_{3j}$ | $c_{3j}$ |
|---|---|---|
| $x_{31}$ | 14 | 11 |
| $x_{32}$ | 18 | 25 |
| $x_{33}$ | 14.99 | 30 |
| $x_{34}$ | 4.99 | 3 |

$capacity = 40$

(a)  (b)  (c)

Figure 3: (a) combined primal graph over 6 variables $\{X_1, X_2, X_3, X_1', X_2', X_3'\}$ and two log-linear models, each having three log-potentials. The functions $f_1'$, $f_2'$ and $f_3'$ form the objective while the functions $g_1$, $g_2$ and $g_3$ form the constraint. (b) defines the CMPE task and (c) shows an equivalent MCKP to the CMPE task given in (b). Solution to the MCKP and the corresponding solution to CMPE is highlighted in red. The weight of the optimal solution is $15.01 + 8 + 4.99 = 28$.

Ihler et al. propose three different classes of message-passing schemes on join graphs. The first scheme which yields the classic mini-buckets (MB) algorithm [11] performs message-passing along a given order $o$ of variables processing the clusters one by one. For each variable $X$ along $o$, the algorithm *maxes out* $X$ from each unprocessed cluster that it appears in and sends a message to all unprocessed clusters that are connected to the cluster but do not mention $X$. The second scheme called *mini-bucket with max-marginal matching* (MM) enhances the MB algorithm by performing LP-tightening updates on all clusters that are processed when maxing-out $X$. These update equations are inspired by work on a family of LP-based methods including reweighted max-product [34], max-product linear programming (MPLP) [14], dual decomposition [21], and soft arc consistency [4, 33]. At a high level, these tightening approaches shift cost/weight from one cluster to another without changing the original weight and thus try to *re-parameterize* the model in order to reduce and thus improve the upper bound. The third scheme, which we refer to as JG, is a fully iterative approach which performs message-passing or LP-tightening updates until convergence or until a bound on the number of iterations is reached. In practice, it was observed that in terms of the upper bound quality, JG is superior to MM which in turn is superior to MB. However, since MM and MB are single pass approaches while JG is iterative, JG requires significantly more time.

Fig. 2 demonstrates the MB and MM algorithms using the log-linear model given in Fig. 1 having log-potentials $f_1$, $f_2$ and $f_3$. MB is a one-pass algorithm and computes the messages denoted by $h$ while MM also passes the marginal matching messages denoted by $c$ (red).

## 2.2 Multi-Choice Knapsack Problem

The knapsack problem is a classic problem in combinatorial optimization which is often stated as follows. Given $n$ items where each item $i$ has a cost $c_i$ and a profit $p_i$, and a knapsack with capacity $q$, select a subset of items such that the total profit is maximized and the total cost does not exceed $q$. For each item $i$, we can associate a Boolean variable $x_i$ which takes value 1 if the item is selected and 0 if it is not and state it as the following optimization problem: $\max_{\boldsymbol{x}} \sum_{i=1}^{n} p_i x_i$ s.t. $\sum_{i=1}^{n} c_i x_i \leq q$.

The *multiple-choice knapsack problem* (MCKP) is a generalization of the knapsack problem in which items are partitioned into groups/bins and the constraint is that exactly one item must be chosen from each group. Specifically, given $m$ bins where each bin $i$ has $N_i$ items and each item $j$ in bin $i$ has a cost $c_{ij}$ and a profit $p_{ij}$, and a knapsack with capacity $q$, the MCKP task is to select one item from each bin such that the total profit is maximized and the total cost does not exceed $q$. For each item $j$ in bin $i$, we can associate a Boolean variable $x_{ij}$ that takes value 1 if the item is selected and 0 if the item is not and express the problem mathematically as:

$$\max_{\boldsymbol{x}} \sum_{i=1}^{m} \sum_{j \in N_i} p_{ij} x_{ij} \text{ s.t. } \sum_{i=1}^{m} \sum_{j \in N_i} c_{ij} x_{ij} \leq q \text{ and } \sum_{j \in N_j} x_{ij} = 1, \ \ i = 1, \ldots, m \qquad (2)$$

In recent work, Rouhani et al. [32] showed that when the combined primal graph of $\mathcal{M}_1$ and $\mathcal{M}_2$ is disconnected into multiple components and the number of variables in each component is bounded by $k$ then the CMPE task can be encoded as MCKP where the number of items in each bin is exponential in $k$. Thus, if the maximum domain size of the variables is $d$, then the number of items in each bin is bounded by $d^k$. Fig. 3 demonstrates the MCKP encoding for an example CMPE task.

If the profits and costs of the items are integers, the MCKP problem can be solved in $O(nq)$ time and $O(mq)$ space where $n$ is the number of items, $m$ is the number of bins and $q$ is the capacity using a dynamic programming algorithm [26, 27, 28]. When the profits and costs are not integers, a LP upper bound can be computed in $O(n)$ time using the Dyer-Zemel algorithm [12, 36].

## 3 Upper Bounding Techniques for CMPE

A straight forward upper bound (LP-Bound) for CMPE can be derived by converting it to an integer linear program, relaxing the integrality constraint to yield a linear program and solving the latter using standard methods and solvers such as the simplex, ellipsoid and interior-point methods. However, the computational complexity of these algorithms is weakly polynomial. To this end, we seek to develop algorithms that are non-trivial and have strong polynomial time guarantees.

We derive two classes of upper bounding algorithms for CMPE. The first class relaxes the global constraint yielding an MPE task while the second duplicates variables by adding equality constraints yielding an MCKP problem. We describe the two approaches in the next two subsections.

### 3.1 MPE based upper bounds by relaxing the global constraint

The *Lagrangian relaxation* of a constrained optimization problem is obtained by incorporating all or some of the constraints into the objective function using appropriate coefficients called the Lagrange multipliers. This relaxation provides a simple and efficient approach to compute upper bounds on the maximization problem known as the *primal* problem. Given a parameter $\lambda \geq 0$ called the Lagrange multiplier, we consider the following Lagrangian relaxation of CMPE:

$$\max_{\boldsymbol{x}} \sum_{f \in \mathcal{F}} f(\boldsymbol{x}) - \lambda \sum_{g \in \mathcal{G}} g(\boldsymbol{x}) + \lambda q \text{ s.t. } \lambda \geq 0 \tag{3}$$

Let $u_\lambda^*$ denote the optimal value of the problem given in Eq. (3). Computing $u_\lambda^*$ can be easily reduced to the MPE task. We construct a set of functions $\mathcal{H}$ as follows. We begin with an empty $\mathcal{H}$. Then, for each $f \in \mathcal{F}$ and its corresponding function $g \in \mathcal{G}$ such that $S(f) = S(g)$ we add a function $h = f - \lambda g$ to $\mathcal{H}$. The MPE value of the log-linear model $\langle \boldsymbol{X}, \mathcal{H} \rangle$ plus $\lambda q$ equals $u_\lambda^*$.

From the theory of Lagrangian relaxations, it is easy to show that $u_\lambda^*$ is an upper bound on $c^*$: all feasible solutions of CMPE are also feasible solutions of the Lagrangian relaxed problem and the objective value of the feasible solutions of CMPE is smaller than or equal to the relaxed problem because $\sum_{g \in \mathcal{G}} g(\boldsymbol{x}) - q \leq 0$. The optimal, namely the smallest upper bound via Lagrangian relaxation is obtained by searching for a value for $\lambda$ s.t. $u_\lambda^*$ is minimized. This minimization problem is called the Lagrangian *dual* problem. Formally, we can show that

**Proposition 1.** $\min_{\lambda:\lambda \geq 0} u_\lambda^* \geq c^*$ *where $u_\lambda^*$ and $c^*$ are optimal values of problems given in Eqs.* (3) *and* (1) *respectively.*

Let $u_{LR}^* = \min_{\lambda:\lambda \geq 0} u_\lambda^*$ denote the optimal upper bound. $u_{LR}^*$ can be computed using standard sub-gradient optimization methods. In particular, $u_\lambda^*$ as a function of $\lambda$ has several nice properties such as concavity but is non-differentiable in presence of multiple optima. To this end, we consider the following update rule for $\lambda$. We start with a heuristic value $\lambda^0$ and generate a sequence $[\lambda^0, \ldots, \lambda^k, \lambda^{k+1}, \ldots]$ using: $\lambda^{k+1} = \lambda^k + \alpha_k \left( \sum_{g \in \mathcal{G}} g(\boldsymbol{x}^{*,k}) - q \right)$ where $\alpha_k$ is the learning rate and $\boldsymbol{x}^{*,k}$ is the optimal solution to Eq. (3) with $\lambda = \lambda^k$.

The discussion above yields Algorithm 1 for computing an upper bound on CMPE. After initializing the variables, specifically the Lagrangian multiplier $\lambda$ and the step-size $\alpha$ to a random positive number and the best answer $u_{LR}^*$ to $\infty$, steps 3 to 10 iteratively solve the Lagrangian dual problem by performing sub-gradient descent over $\lambda$. At each iteration, we construct the log-linear model $\mathcal{M}_\lambda$ (step 3), solve the MPE problem over $\mathcal{M}_\lambda$ to yield an optimal assignment $\boldsymbol{x}_\lambda^*$ (step 4), use $\boldsymbol{x}_\lambda^*$ to

---

**Algorithm 1:** UB-MPE ($\mathcal{M}_1$, $\mathcal{M}_2$, $q$)

---

**Input:** Log-linear models $\mathcal{M}_1 = \langle \mathbf{X}, \mathcal{F} \rangle$ and $\mathcal{M}_2 = \langle \mathbf{X}, \mathcal{G} \rangle$; and a real value $q \in \mathbb{R}$
**Output:** An upper bound on CMPE
**Begin:**
 1: Initialize: (1) $\lambda$ = random number > 0; (2) $u_{LR}^* = \infty$, $\alpha$ = a random number > 0
 2: **repeat**
 3:     Construct $\mathcal{M}_\lambda = \langle \boldsymbol{X}, \mathcal{H} \rangle$ where $\mathcal{H} = \{h = f - \lambda g | f \in \mathcal{F} \text{ and } g \in \mathcal{G}\}$
 4:     Use an MPE algorithm over $\mathcal{M}_\lambda$ to compute the MPE solution $\boldsymbol{x}_\lambda^*$.
        {If MPE is not tractable, we can use algorithms such as MB, MM and JG to upper bound it.}
 5:     $u_\lambda = w_{\mathcal{M}_\lambda}(\boldsymbol{x}_\lambda^*) + \lambda q$
 6:     **if** $u_\lambda < u_{LR}^*$ **then** $u_{LR}^* = u_\lambda$
 7:     $\lambda = \lambda + \alpha \left( \sum_{g \in \mathcal{G}} g(\boldsymbol{x}_\lambda^*) - q \right)$
 8:     Update $\alpha$ using a suitable step-size update procedure (cf. [5])
 9: **until** convergence
10: **return** $u_{LR}^*$

**End.**

---

compute the upper bound $u_\lambda$ (step 5), update the answer $u_{LR}^*$ if the current bound is better (step 6), perform a sub-gradient step to update $\lambda$ (step 7) and update the learning rate using a suitable update procedure (cf. [5]). In our experiments, we used diminishing step size which reduces $\alpha$ by periodically dividing it by a positive constant.

To use Algorithm 1 in practice, we need access to an MPE solver. In particular, if MPE is tractable, for example the treewidth of the combined primal graph is small or the model admits a tractable probabilistic circuit such as a cutset network [29] or an arithmetic circuit [8] and the sub-gradient optimization converges in $t$ iterations, we can show that the bound returned by Algorithm 1 can never be worse than the linear programming relaxation. Formally (proof in the supplement),

**Theorem 1.** *If the MPE over $\mathcal{M}_\lambda$ is tractable for all $\lambda \geq 0$ then $u_{LR}^* \leq u_{LP}^*$, where $u_{LP}^*$ is the LP-bound of CMPE (Eq. (1)) and $u_{LR}^*$ is the upper bound returned by Algorithm 1.*

When $\mathcal{M}_\lambda$ is not tractable, we propose to use polynomial time approximations whose complexity can be controlled using an integer parameter called the $i$-bound; specifically MB, MM and JG methods (see section 2.1) to yield an upper bound on $u_\lambda$ at each iteration. This yields an upper bound that may be worse (higher) than $u_{LR}^*$; however using only polynomial time and space complexity.

### 3.2 MCKP based upper bounds using the Lagrange decomposition method

The main idea in our proposed scheme is to decompose the structures of both $\mathcal{M}_1$ and $\mathcal{M}_2$ by duplicating variables such that the CMPE task reduces to MCKP and then solving the latter using specialized algorithms to yield an upper bound. We describe our idea using the following CMPE problem:

$$\max_{\boldsymbol{x}} f_1(x_1, x_2) + f_2(x_2, x_3) + f_3(x_1, x_3) \text{ s.t. } g_1(x_1, x_2) + g_2(x_2, x_3) + g_3(x_1, x_3) \leq q \quad (4)$$

Here, we assume that all variables are binary and take values from the set $\{0, 1\}$ and $\boldsymbol{x} = (x_1, x_2, x_3)$. Duplicating each variable two times since each variable appears in two functions and adding equality constraints to account for the duplication, we can express the problem given in Eq. (4) as:

$$\max_{\boldsymbol{x}, \boldsymbol{x}'} f_1(x_1', x_2) + f_2(x_2', x_3) + f_3(x_1, x_3') \text{ s.t. } g_1(x_1', x_2) + g_2(x_2', x_3) + g_3(x_1, x_3') \leq q \quad (5)$$

$$\text{and } x_1 = x_1', \ x_2 = x_2' \text{ and } x_3 = x_3' \quad (6)$$

Relaxing the constraints using Lagrange multipliers $\boldsymbol{\mu} = (\mu_1, \mu_2, \mu_3)$, where $\mu_1, \mu_2, \mu_3 \in \mathbb{R}$ we get:

$$\max_{\boldsymbol{x}, \boldsymbol{x}'} f_1(x_1', x_2) + f_2(x_2', x_3) + f_3(x_1, x_3') + \mu_1(x_1 - x_1') + \mu_2(x_2 - x_2') + \mu_3(x_3 - x_3') \quad (7)$$

$$\text{s.t. } g_1(x_1', x_2) + g_2(x_2', x_3) + g_3(x_1, x_3') \leq q \quad (8)$$

Note that by theory of Lagrange relaxations, a solution to the problem described by Eqs. (7) and (8) will yield an upper bound on the problem described by Eqs. (5) and (6). Let $f_1' = f_1 - \mu_1 x_1' + \mu_2 x_2$,

**Algorithm 2:** UB-KP ($\mathcal{M}_1$, $\mathcal{M}_2$, $q$, $i$)

---

**Input:** Log-linear models $\mathcal{M}_1 = \langle \mathbf{X}, \mathcal{F} \rangle$ and $\mathcal{M}_2 = \langle \mathbf{X}, \mathcal{G} \rangle$; a real value $q \in \mathbb{R}$; $i$-bound$\in \mathbb{N}$

**Output:** An upper bound on CMPE

**Begin:**

1: Merge (via summation) functions in $\mathcal{F}$ (and corresponding ones in $\mathcal{G}$) such that the number of variables in the scope of the new functions is bounded by $i$ to yield a new set of functions $\mathcal{H}$ (and $\mathcal{S}$ for $\mathcal{G}$).

2: Given $X_j \in \mathbf{X}$, let $I_j = \{a | h_a \in \mathcal{H}, s_a \in \mathcal{S}$ and $X_j \in S(h_a) = S(s_a)\}$ and $\{Y_{j,a}\}$ where $a \in I_j$ denote the set of duplicated variables of $X_j$. Let $\mathbf{Y} = \{Y_{j,a} | X_j \in \mathbf{X}$ and $a \in I_j\}$

3: Express $\mathcal{H}$ and $\mathcal{S}$ using duplicated variables $\mathbf{Y}$

4: Initialize: (1) all members of vector $\boldsymbol{\mu} = \{\mu_{j,a,b} | X_j \in \mathbf{X}, a, b \in I_j,$ and $a < b\}$ to a random number; (2) $u_{LD}^* = \infty$, and (3) $\alpha =$ a random number $> 0$

5: **repeat**

6:     Construct a MCKP from $\mathcal{H}$, $\mathcal{S}$, $\boldsymbol{\mu}$ and $q$

7:     Solve the MCKP to yield an optimal value $u_{\boldsymbol{\mu}}^*$ and an assignment of values $\boldsymbol{y}^*$ to $\mathbf{Y}$.
    {If MCKP is intractable, we solve it approximately using a linear time upper bounding schemes [19].}

8:     **if** $u_{\boldsymbol{\mu}}^* < u_{LD}^*$ **then** $u_{LD}^* = u_{\boldsymbol{\mu}}^*$

9:     Update each $\mu_{j,a,b}$ using $\mu_{j,a,b} - \alpha(y_{j,a}^* - y_{j,b}^*)$

10:    Update $\alpha$ using a suitable step-size update procedure (cf. [5])

11: **until** convergence

12: **return** $u_{LD}^*$

**End.**

---

$f_2' = f_2 - \mu_2 x_2' + \mu_3 x_3$ and $f_3' = f_3 - \mu_3 x_3' + \mu_1 x_1$. Then we can express the problem described by Eqs. (7) and (8) as:

$$\max_{\boldsymbol{x}, \boldsymbol{x}'} f_1'(x_1', x_2) + f_2'(x_2', x_3) + f_3'(x_1, x_3') \text{ s.t. } g_1(x_1', x_2) + g_2(x_2', x_3) + g_3(x_1, x_3') \leq q \qquad (9)$$

The problem described by Eq. (9) is an instance of MCKP having 3 bins and 4 items in each bin since the combined primal graph associated with the CMPE problem has three connected components corresponding to the three functions (see Fig. 3). Thus, it can be solved using MCKP methods to yield an upper bound on our example CMPE problem. Let $u_{\boldsymbol{\mu}}^*$ denote the optimal value of the problem described by Eq. (9), then solving the Lagrangian dual $u_{LD}^* = \min_{\boldsymbol{\mu}} u_{\boldsymbol{\mu}}^*$ via sub-gradient optimization methods yields the best possible upper bound.

**Example 2.** *The combined primal graph given in Fig. 3 is obtained by duplicating each variable in the combined primal graph of the log-linear models given in Fig. 1 and relaxing the equality constraint (via Lagrangian decomposition) between the duplicated variables such that the number of nodes in each connected component is bounded by 2. Given $\mu_1 = -5.01$, $\mu_2 = -1.0$ and $\mu_3 = 1.0$, the functions $f_1'$, $f_2'$ and $f_3'$ given in Fig. 3(b) are obtained from the functions $f_1$, $f_2$ and $f_3$ respectively given in Fig. 1 using the expressions $f_1' = f_1 - \mu_1 x_1' + \mu_2 x_2$, $f_2' = f_2 - \mu_2 x_2' + \mu_3 x_3$ and $f_3' = f_3 - \mu_3 x_3' + \mu_1 x_1$. The functions $g_1$, $g_2$ and $g_3$ are copied from Fig. 1. Thus, from Eq. (9), theory of Lagrange relaxations and MCKP encoding described in Rouhani et al. [32], solving the MCKP task in Fig. 3(c) will yield an upper bound on the optimal value of the CMPE problem. The value of the optimal solution to MCKP given in Fig. 3(c) is $28$. Thus, for this toy example, the MCKP based upper bound is equal to the optimal value of the CMPE problem.*

We generalize the method prescribed in the aforementioned example to yield a polynomial time upper bounding scheme. In particular, in order to control both the computational time and space complexity of our proposed algorithm, we use an integer parameter $i$ (similar to the $i$-bound used for join graphs). This parameter exponentially bounds the number of items in each bin of MCKP and under the assumption that the log-potentials have integer values that are bounded above by an integer $B$, the MCKP generated by our relaxation can be solved in time that scales linearly in $\exp(i)$, $B$ and the number of functions using a dynamic programming algorithm. Before we present our algorithm, we introduce some required notation. Given a set of functions $\mathcal{H}$ that defines the objective, corresponding functions $\mathcal{S}$ that define the constraint and a variable $X_j \in \mathbf{X}$, let $I_j = \{a | h_a \in \mathcal{H}, s_a \in \mathcal{S}$ and $X_j \in S(h_a) = S(s_a)\}$ denote an index over functions that mention $X_j$. Let $\{Y_{j,a}\}$ where $a \in I_j$ denote the set of duplicated variables of $X_j$. Let $\{\mu_{j,a,b}\}$ denote the set of Lagrange multipliers associated with each variable $X_j \in \mathbf{X}$ where $a, b \in I_j$ and $a < b$. Without loss of generality, we can now state the Lagrangian relaxed problem as:

$$\max_{\boldsymbol{y}} \sum_{h \in \mathcal{H}} h(\boldsymbol{y}) + \sum_{j,a,b | a,b \in I_j, a<b} \mu_{j,a,b}(y_{j,a} - y_{j,b}) \text{ s.t. } \sum_{s \in \mathcal{S}} s(\boldsymbol{y}) \leq q$$

Table 1: Table showing results on cutset networks for 4 datasets `AD`, `BBC`, `Book` and `Reuters`. The line under each network reports the number of random variables and their maximum domain size. Alg: Algorithm, q20 and q50: q values are chosen from the 20-th and 50-th percentile respectively, and Tm: time in seconds.

| | AD (1556,2) | | | | BBC (1058,2) | | | | Book (500,2) | | | | Reuters (889,2) | | | |
|------|---------|----|---------|----|--------|----|--------|----|--------|----|--------|----|---------|----|---------|-----|
| Alg | q20 | Tm | q50 | Tm | q20 | Tm | q50 | Tm | q20 | Tm | q50 | Tm | q20 | Tm | q50 | Tm |
| MB | **-3308.3** | 0 | **-3228.4** | 0 | **-953.9** | 1 | **-906.7** | 0 | **-670.4** | 0 | **-629.9** | 0 | **-1572.3** | 0 | **-1443.5** | 0 |
| KP | -3307.8 | 59 | -3225.8 | 70 | -953.4 | 42 | -906.1 | 49 | -669.9 | 15 | -629.7 | 20 | -1564.8 | 63 | -1431.6 | 149 |

where $\boldsymbol{Y}$ is the set of duplicated variables. To solve the Lagrangian dual, we can use the sub-gradient method with the following update rule for each $\mu_{j,a,b}$. We start with $\mu_{j,a,b}^0$ and generate a sequence $[\mu_{j,a,b}^0, \mu_{j,a,b}^1, \ldots]$ using: $\mu_{j,a,b}^{k+1} = \mu_{j,a,b}^k - \alpha_k(y_{j,a} - y_{j,b})$ where $\alpha_k$ is the learning rate.

Algorithm 2 formally describes our approach. In step 1, it merges functions in $\mathcal{M}_1$ (and $\mathcal{M}_2$) creating new functions having potentially higher scope sizes under the constraint that the maximum scope size is bounded by $i$. This yields two new sets of functions $\mathcal{H}$ and $\mathcal{S}$ which represent the objective and the constraint respectively. Then the algorithm duplicates the variables so that the resulting problem can be converted to MCKP (step 2-3). In steps 4-12 the algorithm solves the Lagrangian dual problem via sub-gradient descent over the Lagrange multipliers $\boldsymbol{\mu}$. A key sub-step in optimizing the dual is solving the MCKP (step 7). For this, the algorithm either uses an exact MCKP algorithm if feasible or a linear time upper bounding scheme [12, 36].

In general, when the combined primal graph has large number of disconnected components, the quality of bounds returned by Algorithm 2 is likely to be better than the quality of bounds returned by Algorithm 1. For instance, in the extreme case when the primal graph has no edges, Algorithm 2 will either yield an exact answer or superior knapsack-based upper bounds while Algorithm 1 will yield inferior bounds that are equal to the LP-relaxation of the knapsack problem (see [19]).

## 4 Experiments

We compared the upper bounding schemes proposed in this paper on three types of CMPE problems having different levels of complexity. The first type uses tractable probabilistic circuits, specifically cutset networks [29, 30] learned on well known benchmarks used by the tractable models community [23]. The second type uses intractable, high treewidth models from the UAI competitions [13, 15]. The third type uses models developed for performing adversarial attacks on classifiers in order to measure their robustness. We also evaluated the impact of increasing the $i$-bound on the quality of upper bound and the time required to compute the upper bound.

We evaluated three algorithms within our MPE-based bounding approach: (1) Mini-Bucket elimination (MB); (2) Mini-Bucket with max-marginal matching (MM); and (3) join graph based linear programming (JG). We implemented the Dyer-Zemel algorithm [12, 36] to compute upper bounds on MCKP (we will call it KP for brevity). We ran JG for a maximum of 40 iterations or until convergence. We ran the outer-loop of the MPE-based as well as MCKP-based bounding algorithms for a maximum of 100 iterations or until convergence. Details are described in the supplement.

**Results on MPE Tractable Models.** We learned cutset networks on four high-dimensional datasets `AD`, `BBC`, `Book` and `Reuters`. We used these networks as $\mathcal{M}_1$. To construct $\mathcal{M}_2$, we modified the parameters of $\mathcal{M}_1$ using a noise parameter $\epsilon \sim N(0, \sigma^2 = 0.1)$. For each network, we experimented with 5 values of $q$ such that the chosen values lie roughly in the 10-th, 20-th, 50-th, 80-th and 90-th percentile respectively. This helps us evaluate the impact of $q$ on the bounds.

Table 1 shows the results for two values of $q$ (20-th and 50-th percentile). Results for other values of $q$ are included in the supplement. Note that on these networks, the mini-buckets algorithm yields exact MPE values in the inner loop of our algorithm and as a result max-marginal as well as join-graph based propagation will not improve the bounds (we have therefore not reported them). As expected, we see that when MPE is tractable, MPE based bounds are superior in terms of time as well as quality than MCKP based bounds.

Table 2: Table showing results on four networks from the UAI competitions: Segmentation, Grids, Promedas and Pedigree. The line under each network reports the number of random variables, the maximum domain size of the variables, the number of functions and the treewidth. Alg: Algorithm, iB: i-bound, q20 and q50: q values are chosen from the 20th and 50th percentile respectively, and Tm: time in seconds.

| | | Segmentation (232,2,863,18) | | | | Grids (1600,2,4800,113) | | | | Promedas (1953,2,1953,148) | | | | Pedigree (1152,5,1152,35) | | | |
|-----|----|--------|----|--------|----|--------|-----|--------|-----|---------|-----|---------|-----|--------|-----|--------|-----|
| Alg | iB | q20 | Tm | q50 | Tm | q20 | Tm | q50 | Tm | q20 | Tm | q50 | Tm | q20 | Tm | q50 | Tm |
| MB | 2 | -381.7 | 0 | -366.0 | 0 | 1370.4 | 1 | 1587.4 | 0 | -1491.7 | 1 | -1431.5 | 1 | -123.4 | 0 | -111.2 | 0 |
| MM | 2 | -427.3 | 0 | -416.9 | 0 | 1040.8 | 1 | 1235.8 | 1 | -1964.8 | 1 | -1881.8 | 1 | -235.6 | 1 | -222.0 | 1 |
| JG | 2 | **-459.9** | 5 | **-447.8** | 7 | 821.9 | 35 | **1017.8** | 30 | -2466.1 | 45 | -2362.5 | 56 | -319.2 | 37 | -307.4 | 33 |
| KP | 2 | -459.9 | 37 | -446.8 | 36 | **821.4** | 430 | 1020.4 | 406 | **-2765.3** | 139 | **-2677.6** | 138 | **-323.6** | 214 | **-310.9** | 204 |
| MB | 5 | -425.0 | 1 | -409.8 | 0 | 1078.4 | 2 | 1289.6 | 2 | -2661.6 | 1 | -2578.1 | 0 | -268.9 | 1 | -260.2 | 1 |
| MM | 5 | -452.2 | 0 | -440.0 | 1 | 857.3 | 2 | 1045.9 | 2 | -2748.8 | 1 | -2656.7 | 2 | -315.3 | 1 | -302.6 | 1 |
| JG | 5 | **-461.5** | 8 | **-449.1** | 6 | **819.9** | 51 | **1016.0** | 51 | -2764.6 | 37 | -2676.2 | 42 | **-330.8** | 32 | **-318.2** | 25 |
| KP | 5 | -459.8 | 37 | -446.8 | 31 | 819.9 | 214 | 1016.3 | 164 | **-2770.8** | 116 | **-2681.2** | 150 | -324.8 | 139 | -312.6 | 134 |
| MB | 8 | -448.0 | 1 | -434.4 | 0 | 995.8 | 3 | 1178.7 | 3 | -2692.0 | 2 | -2599.4 | 1 | -292.3 | 1 | -285.9 | 1 |
| MM | 8 | -460.2 | 1 | -448.5 | 1 | 830.3 | 6 | 1024.8 | 6 | -2762.7 | 3 | -2670.1 | 4 | -327.6 | 3 | -313.9 | 3 |
| JG | 8 | **-461.1** | 22 | **-449.2** | 25 | 820.3 | 190 | **1016.0** | 160 | -2760.8 | 98 | -2670.7 | 126 | **-327.8** | 91 | **-317.1** | 84 |
| KP | 8 | -459.8 | 29 | -448.1 | 37 | **819.6** | 202 | 1016.0 | 167 | **-2771.4** | 134 | **-2681.7** | 130 | -325.8 | 171 | -313.6 | 170 |
| MB | 10 | -452.7 | 2 | -438.7 | 2 | 951.7 | 8 | 1143.4 | 8 | -2688.4 | 4 | -2601.5 | 4 | -300.9 | 3 | -284.9 | 4 |
| MM | 10 | -460.7 | 2 | -448.5 | 3 | 824.1 | 12 | 1021.2 | 13 | -2763.1 | 8 | -2673.2 | 8 | -330.1 | 6 | **-316.4** | 8 |
| JG | 10 | **-461.4** | 65 | **-448.9** | 68 | 823.9 | 396 | 1018.9 | 501 | -2756.4 | 322 | -2669.9 | 381 | **-331.2** | 288 | -316.4 | 319 |
| KP | 10 | -460.0 | 33 | -448.4 | 36 | **819.5** | 215 | **1016.0** | 170 | **-2770.8** | 115 | **-2680.3** | 101 | -324.9 | 211 | -313.8 | 193 |

**Results on Networks from the UAI Competition.** We also experimented with several high treewidth networks used in the UAI competitions [13, 15]. We varied the $i$-bound of all algorithms from 2 to 10. We used the technique described above for MPE tractable models to construct the CMPE problems. For brevity, in Table 2 we present results on the largest network from four domains: image segmentation, Grids, medical diagnosis (Promedas) and pedigrees for genetic linkage analysis for two values of $q$ which are roughly equal to the 20-th and 50-th percentile respectively.

We see that KP and JG yield smaller (better) upper bounds than MB and MM. However, their time complexity is much higher. For smaller $i$-bounds ($= 2, 5$), there is an order of magnitude difference between the upper bounds output by JG and KP, and the other two approaches. However, the difference is much smaller for larger $i$-bounds ($= 8, 10$). The MM algorithm yields the best trade-off between time complexity and upper bound quality, especially when using higher $i$-bounds is feasible. However, if using higher $i$-bounds in not feasible, the KP algorithm should be used.

**Results on Adversarial Modification on the MNIST dataset.** We consider the problem of performing adversarial attacks on discriminative classifiers in order to measure their robustness. The goal is to manipulate the predictions made by the classifier by minimally changing the test example. We show that this problem can be reduced to CMPE if the classifier can be expressed as a log-linear model. Formally, let $\mathcal{G}$ be a set of features/functions, each defined over a set of random variables $\boldsymbol{X}$. Given an assignment $\boldsymbol{x}$, let $D$ be a decision variable which takes the value $d$ if $\sum_{g \in \mathcal{G}} g(\boldsymbol{x}) > 0$ and $\bar{d}$ otherwise. Given an assignment (test example) $\boldsymbol{x}$, our goal in rendering an adversarial attack is to modify it minimally such that the classifier decision flips, namely find an assignment $\boldsymbol{x}'$ such that the decision flips from $d$ to $\bar{d}$ and the distance between $\boldsymbol{x}$ and $\boldsymbol{x}'$ is minimized. Given an assignment $\boldsymbol{x}$, we can model the complement of the Hamming distance using the set of $n$ univariate functions $\mathcal{F} = \{f_1, \ldots, f_n\}$ where $f_i(x_i') = 1$ if $x_i' \in \boldsymbol{x}$ and 0 otherwise. Notice that if the hamming distance between $\boldsymbol{x}$ and $\boldsymbol{x}'$ is $k$ then $\sum_{i=1}^{n} f_i(x_i') = n - k$. Given an assignment $\boldsymbol{x}$ such that $\sum_{g \in \mathcal{G}} g(\boldsymbol{x}) > 0$ and set of functions $\mathcal{F}$ and $\mathcal{G}$ as described above, we can now express the adversarial attack task as the following CMPE problem: $\max_{\boldsymbol{x}'} \sum_{f \in \mathcal{F}} f(\boldsymbol{x}')$ $s.t.$ $\sum_{g \in \mathcal{G}} g(\boldsymbol{x}') \leq 0$.

We can also use more sophisticated distance functions that take into account the *spatial arrangement* of pixels. For example, if we want pixels that are spatially close to take the same value, we can add the following set of pairwise functions to $\mathcal{F}$: $f_{i,j}(x_i, x_j) = 1.0$ if $X_i$ is a neighboring pixel of $X_j$ and $x_i = x_j$, and 0 otherwise. This will yield a graphical model whose primal graph is a grid.

Table 3: Table showing the average upper bound over 21000 test examples and the time required by the four algorithms MB, MM, JG and KP with i-bounds varying between 2 and 4 on the MNIST dataset using the Grid model for measuring distance and linear SVM as the classifier. Under the univariate distance model, the average upper bounds for MB and KP with i-bound set to 1 were 781.05±2.662 and **780.8**±2.926 respectively. Avg. UB: average upper bound, Std. UB: standard deviation of upper bound and Avg. Time: average time in seconds.

| | i-bound=2 | | | | i-bound=3 | | | | i-bound=4 | | | |
| | MB | MM | JG | KP | MB | MM | JG | KP | MB | MM | JG | KP |
|---|---|---|---|---|---|---|---|---|---|---|---|---|
| Avg. UB | 291.42 | 290.91 | **290.11** | 290.85 | 291.0 | 290.38 | **290.11** | 290.69 | 290.79 | 290.38 | **290.11** | 290.66 |
| Std. UB | 2.12 | 2.14 | 2.44 | 2.32 | 2.15 | 2.35 | 2.42 | 2.33 | 2.2 | 2.36 | 2.42 | 2.38 |
| Avg. Time | 0.0 | 0.0 | 2.8 | 0.6 | 0.0 | 0.0 | 3.0 | 0.4 | 0.0 | 0.0 | 3.3 | 0.3 |

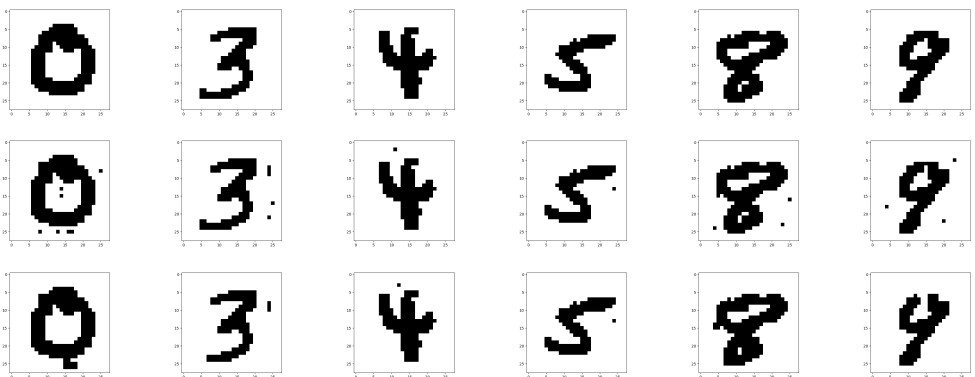

Figure 4: Qualitative results on MNIST. Row 1 shows the original test images. Row 2 shows the images changed using the univariate model *so that the decision changes* from the original classification to any one of the other 9 digits. Row 3 shows the images changed using the grid model.

We conducted experiments on the MNIST dataset [22] using the univariate and grid distance functions described above and linear support vector machine as the classifier. The dataset has 70,000 examples. We used 70% of the examples for training and 30% for testing. The accuracy of our classifier on the dataset was 97%. Note that a limitation of our approach in its current form is that we can only use multi-linear classifiers (aka graphical models without latent variables) and cannot use non-linear classifiers such as deep neural networks (however, we can use their linear or high treewidth approximations and this is a subject of future research).

Table 3 shows quantitative results comparing the quality of the upper bounds and time required for the univariate and grid distance functions. We see that on the univariate model, the KP based bounds yield exact answers (since the univariate problem is an instance of the knapsack problem) while the MPE based bounds are only slightly worse than the KP based bounds. On the grid models however, the MPE bounds are more accurate.

Fig. 4 shows qualitative results. The qualitative results verify our intuition that the grid model yields smoother deceptions than the univariate model. More qualitative results on the MNIST dataset are included in the supplement.

## 5  Conclusion

We proposed novel upper bounding methods for solving the CMPE task. The schemes are based on relaxing the original constrained maximization problem into either an MPE or MCKP problem which can then be solved using state-of-the-art techniques. Our empirical findings on a large variety of models including both tractable and intractable models suggest that our proposed relaxations can produce effective upper bounds. We also presented the application of CMPE in solving an important task in robust estimation: measuring the robustness of discriminative classifiers. On the MNIST dataset we showed that CMPE equipped with our proposed upper bounding methods can efficiently find the most important $k$ pixels to change the prediction of the classifier. In future, we will explore other interesting applications of CMPE involving multiple constraints and over non-linear models.

## Acknowledgements

This work was supported in part by the DARPA Explainable Artificial Intelligence (XAI) Program under contract number N66001-17-2-4032, and by the National Science Foundation grants IIS-1652835 and IIS-1528037.

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
