# Supplement:
# Novel Upper Bounds for the Constrained Most Probable Explanation Task

**Tahrima Rahman**     **Sara Rouhani**     **Vibhav Gogate**
The University of Texas at Dallas
{tahrima.rahman, sara.rouhani, vibhav.gogate@utdallas.edu}

## A   Proof of Theorem 1

*Proof.* It is well known that any MPE task can be encoded as an integer linear programming (ILP) problem (cf. [4]). A popular or widely used formulation is to associate a Boolean variable with each entry in each function of the log-linear model. When the Boolean variable is assigned the value 1, the entry is selected, otherwise it is not. With these Boolean variables, the objective function can be written as: $\sum_i w_i y_i$ where $w_i$ is the entry in a function and $y_i \in \{0, 1\}$ is the value assigned to the corresponding Boolean variable $Y_i$. The constraints in the ILP encode various consistency constraints which ensure that only entries which yield non-conflicting or consistent assignment to all variables in the original MPE problem are selected. For instance, a type of consistency constraint encodes the restriction that only entry from each function must be selected. A second type of consistency constraint ensures that if two functions share a variable then only entries which assign the shared variable to the same value are selected. For example, given functions $f_1(x_1.x_2)$ and $f_2(x_2, x_3)$ which share the variable $X_2$, the consistency constraint ensures that conflicting entries such as $(x_1 = 0, x_2 = 1)$ and $(x_2 = 0, x_3 = 1)$ which differ on the value assigned to $x_2$ are not selected. In other words, given a Markov network $\mathcal{M}$, we can express the MPE problem over it using an ILP of the following form:

$$\max \sum_i w_i y_i$$
$$\text{s.t. } \sum_{i=1} a_{ij} y_i \leq c_j; \ j = 1, \ldots, m \qquad (1)$$
$$y_i \in \{0, 1\}$$

where $\sum_{i=1} a_{ij} y_i \leq c_j; \ j = 1, \ldots, m$ are the consistency constraints and $a_{ij}, c_j \in \mathbb{R}$ for all $i, j$.

The CMPE task adds a global constraint to the ILP formulation of MPE given in Eq. (1). Let $v_i$ denote the entries associated with functions in $\mathcal{M}_2$. Then, we can encode the constraint in CMPE as $\sum_i v_i y_i \leq q$. This gives us the following ILP formulation of CMPE:

$$\max \sum_i w_i y_i$$
$$\text{s.t. } \sum_i v_i y_i \leq q$$
$$\sum_{i=1} a_{ij} y_i \leq c_j; \ j = 1, \ldots, m \qquad (2)$$
$$y_i \in \{0, 1\}$$

35th Conference on Neural Information Processing Systems (NeurIPS 2021).

The LP-bound on the ILP given in Eq. (2) is obtained by relaxing the integrality requirement $y_i \in \{0, 1\}$ to $0 \leq y_i \leq 1$. This gives us the following linear programming relaxation.

$$\max \sum_i w_i y_i$$
$$\text{s.t. } \sum_i v_i y_i \leq q \tag{3}$$
$$\sum_{i=1} a_{ij} y_i \leq c_j; \ j = 1, \ldots, m$$
$$0 \leq y_i \leq 1$$

Let $c^*$ and $u_{LP}^*$ denote the optimal value of the ILP and LP given in Eqs. (2) and (3). Since all feasible solutions of the ILP are also feasible solutions of the LP, it follows that $u_{LP}^* \geq c^*$.

Let us consider a Lagrange relaxation of the ILP given in Eq. (2) which is obtained by relaxing the constraint $\sum_i v_i y_i \leq q$ using a Lagrange multiplier $\lambda$. This gives us the following ILP:

$$\max \left\{ \lambda q + \sum_i (w_i - \lambda v_i) y_i \right\}$$
$$\text{s.t. } \sum_{i=1} a_{ij} y_i \leq c_j; \ j = 1, \ldots, m \tag{4}$$
$$y_i \in \{0, 1\}$$
$$\lambda \geq 0$$

Let $v_\lambda^*$ denote the optimal value of the problem given in Eq. (4) for a given value of $\lambda$. This problem is called the Lagrangian dual and it is known that for any value $\lambda \geq 0$, $v_\lambda^* \geq c^*$. Thus, the optimal value of the dual, namely $v_{LR}^* = \min_\lambda v_\lambda^*$ is also an upper bound on $c^*$. From Lagrange relaxation and duality theory (cf. [7, 10]), it is well known that

$$v_{LR}^* \leq u_{LP}^* \tag{5}$$

Comparing the ILP problems given in Eqs. (1) and (4), it is easy to see that the latter is an instance of the MPE problem. This MPE problem is the same problem solved at each iteration in Algorithm 1 (see step 4 in the algorithm described in the main paper). Thus, under the assumption that MPE is solved exactly at each iteration, namely MPE is tractable, the output of Algorithm 1 (see the main paper) equals $v_{LR}^*$, namely $u_{LR}^* = v_{LR}^*$. Substituting $u_{LR}^* = v_{LR}^*$ in Eq. (5), we have $c^* \leq u_{LR}^* \leq u_{LP}^*$.

$\square$

## B  Experiments

We compared the upper bounding schemes proposed in this paper on three types of CMPE problems having different levels of complexity. The first type uses tractable probabilistic circuits, specifically cutset networks [8, 9] learned on well known benchmarks used by the tractable models community [6]. The second type uses intractable, high treewidth models from the UAI competitions [2, 3]. The third type uses models developed for performing adversarial attacks on classifiers in order to measure their robustness. We also evaluated the impact of increasing the $i$-bound on the quality of upper bound and the time required to compute the upper bound.

We evaluated three algorithms within our MPE-based bounding approach: (1) Mini-Bucket elimination (MB); (2) Mini-Bucket with max-marginal matching (MM); and (3) join graph based linear programming (JG). We implemented the Dyer-Zemel algorithm [1, 11] to compute upper bounds on MCKP (we will call it KP for brevity). We ran JG for a maximum of 40 iterations or until convergence. We ran the outer-loop of the MPE-based as well as MCKP-based bounding algorithms for a maximum of 100 iterations or until convergence. Details are described in the supplement.

**Results on MPE Tractable Models.**   We learned cutset networks on four high-dimensional datasets AD, BBC, Book and Reuters. We used these networks as $\mathcal{M}_1$. To construct $\mathcal{M}_2$, we modified the parameters of $\mathcal{M}_1$ using a noise parameter $\epsilon \sim N(0, \sigma^2 = 0.1)$. For each network, we experimented

Table 1: $q$ values used in our experiments. The numbers in the bracket along side each network reports the number of random variables, the maximum domain size of the variables, the number of functions and the treewidth (computed using the min-fill heuristic) of the network.

| Network | q10 | q20 | q50 | q80 | q90 |
|---|---|---|---|---|---|
| MPE Tractable Models | | | | | |
| Reuters (889,2,889,1) | -2370.16 | -1945.39 | -1820.15 | -1274.92 | -1151.26 |
| Book (500,2,500,1) | -1126.96 | -874.55 | -832.91 | -797.14 | -755.12 |
| AD (1556,2,1556,1) | -4420.69 | -3974.46 | -3893.01 | -3824.22 | -3743.26 |
| BBC (1058,2,1058,1) | -1661.27 | -1389.32 | -1338.56 | -1295.27 | -1244.34 |
| High Treewidth Models | | | | | |
| Medical Diagnosis (1953,2,1953,148) | -4278.73 | -3754.09 | -3664.41 | -3587.37 | -3498.1 |
| 40x40 Grids (1600,2,4800,113) | -1537.85 | -276.31 | -79.05 | 89.59 | 286.06 |
| Image Segmentation (232,2,863,18) | -720.67 | -640.42 | -625.94 | -613.21 | -597.8 |
| Linkage Analysis (1152,5,1152,35) | -897.26 | -797.09 | -779.54 | -764.27 | -746.75 |

Table 2: Table showing results on a MPE tractable cutset network [9] learned from the `AD` dataset. The two numbers in brackets on the first line report the number of random variables and their maximum domain size. Alg: Algorithm, q10, q20, q50, q80 and q90: q values are chosen from the 10th, 20th, 50th, 80th and 90th percentile respectively, and Tm: time in seconds.

| | AD (1556,2) | | | | | | | | | |
|---|---|---|---|---|---|---|---|---|---|---|
| Alg | q10 | Tm | q20 | Tm | q50 | Tm | q80 | Tm | q90 | Tm |
| MB | **-3750.9** | 0 | **-3308.3** | 0 | **-3228.4** | 0 | **-3161.2** | 0 | **-3082.5** | 0 |
| KP | -3749.0 | 46 | -3307.8 | 59 | -3225.8 | 70 | -3159.2 | 58 | -3081.6 | 62 |

Table 3: Table showing results on a MPE tractable cutset network [9] learned from the `BBC` dataset. The two numbers in brackets on the first line report the number of random variables and their maximum domain size. Alg: Algorithm, q10, q20, q50, q80 and q90: q values are chosen from the 10th, 20th, 50th, 80th and 90th percentile respectively, and Tm: time in seconds.

| | BBC (1058,2) | | | | | | | | | |
|---|---|---|---|---|---|---|---|---|---|---|
| Alg | q10 | Tm | q20 | Tm | q50 | Tm | q80 | Tm | q90 | Tm |
| MB | **-1213.7** | 0 | **-953.9** | 1 | **-906.7** | 0 | **-866.6** | 0 | **-819.7** | 0 |
| KP | -1213.6 | 52 | -953.4 | 42 | -906.1 | 49 | -865.8 | 71 | -819.3 | 53 |

Table 4: Table showing results on a MPE tractable cutset network [9] learned from the `Book` dataset. The two numbers in brackets on the first line report the number of random variables and their maximum domain size. Alg: Algorithm, q10, q20, q50, q80 and q90: q values are chosen from the 10th, 20th, 50th, 80th and 90th percentile respectively, and Tm: time in seconds.

| | Book (500,2) | | | | | | | | | |
|---|---|---|---|---|---|---|---|---|---|---|
| Alg | q10 | Tm | q20 | Tm | q50 | Tm | q80 | Tm | q90 | Tm |
| MB | **-922.7** | 0 | **-670.4** | 0 | **-629.9** | 0 | **-595.2** | 0 | **-554.8** | 0 |
| KP | -922.1 | 16 | -669.9 | 15 | -629.7 | 20 | -594.9 | 14 | -554.0 | 20 |

Table 5: Table showing results on a MPE tractable cutset network [9] learned from the `Reuters` dataset. The two numbers in brackets on the first line report the number of random variables and their maximum domain size. Alg: Algorithm, q10, q20, q50, q80 and q90: q values are chosen from the 10th, 20th, 50th, 80th and 90th percentile respectively, and Tm: time in seconds.

| | Reuters (889,2) | | | | | | | | | |
|---|---|---|---|---|---|---|---|---|---|---|
| Alg | q10 | Tm | q20 | Tm | q50 | Tm | q80 | Tm | q90 | Tm |
| MB | **-2009.0** | 1 | **-1572.3** | 0 | **-1443.5** | 0 | **-899.7** | 0 | **-781.2** | 1 |
| KP | -2001.7 | 284 | -1564.8 | 63 | -1431.6 | 149 | -897.0 | 215 | -779.5 | 174 |

with 5 values of $q$ such that the chosen values lie roughly in the 10-th, 20-th, 50-th, 80-th and 90-th percentile respectively. This helps us evaluate the impact of $q$ on the bounds. Table 1 shows the values of $q$ used in our experiments for the MPE tractable as well as high treewidth models.

Tables 2, 3, 4 and 5 show the results. Note that on these networks, the mini-buckets algorithm (MB) yields exact MPE values in the inner loop of our algorithm and as a result max-marginal as well as join-graph based propagation will not improve the bounds (we have therefore not reported them). As expected, we see that when MPE is tractable, MPE based bounds are superior in terms of time as well as quality than MCKP based bounds.

**Results on Networks from the UAI Competition.** We also experimented with several high treewidth networks used in the UAI competitions [2, 3]. We varied the $i$-bound of all algorithms from 2 to 10. We used the technique described above for MPE tractable models to construct the CMPE problems. For brevity, in Tables 6, 7, 8 and 9 we present results on the largest networks from four domains: image segmentation, Grids, medical diagnosis (Promedas) and pedigrees for genetic linkage analysis.

We see that KP and JG yield smaller (better) upper bounds than MB and MM. However, their time complexity is much higher. For smaller $i$-bounds ($= 2, 5$), there is an order of magnitude difference between the upper bounds output by JG and KP, and the other two approaches. However, the difference is much smaller for larger $i$-bounds ($= 8, 10$). The MM algorithm yields the best trade-off between time complexity and upper bound quality, especially when using higher $i$-bounds is feasible. However, if using higher $i$-bounds in not feasible, the KP algorithm should be used.

Figures 1, 2, 3 and 4 show how the upper bound computed by MB, MM, JG and KP changes as a function of time for 4 different $i$-bounds: 2, 5, 8 and 10. For each $i$-bound, we chose the value of $q$ randomly. We see that the MPE-based schemes, because they optimize just one parameter $\lambda$ converge quickly as compared to the MCKP-based schemes. These plots suggest that KP will benefit greatly from using advanced sub-gradient descent methods such as taking sub-optimal descent steps where the idea is to sub-optimally solve the MCKP at each iteration until a reasonable assignment for the Lagrange multipliers is found. Then, with this assignment as the starting point, the algorithm proceeds as usual, taking optimal steps until convergence.

**Results on Adversarial Modification on the MNIST dataset.** We consider the problem of performing adversarial attacks on discriminative classifiers in order to measure their robustness. The goal is to manipulate the predictions made by the classifier by minimally changing the test example. We show that this problem can be reduced to CMPE if the classifier can be expressed as a log-linear model. Formally, let $\mathcal{G}$ be a set of features/functions, each defined over a set of random variables $\boldsymbol{X}$.

Table 6: Table showing results on the Image Segmentation instance used in the UAI competitions. The line under the network reports the number of random variables, the maximum domain size of the variables, the number of functions and the treewidth. Alg: Algorithm, iB: i-bound, q10, q20, q50, q80 and q90: q values are chosen from the 10th, 20th, 50th, 80th and 90th percentile respectively, and Tm: time in seconds. The standard deviation for all methods was close to zero and therefore not reported.

| Alg | iB | q10 | Tm | q20 | Tm | q50 | Tm | q80 | Tm | q90 | Tm |
|-----|----|-----|----|-----|----|-----|----|-----|----|-----|----|
| | | | | | Segmentation (232,2,863,18) | | | | | | |
| MB | 2 | -445.4 | 0 | -381.7 | 0 | -366.0 | 0 | -357.8 | 0 | -342.2 | 0 |
| MM | 2 | -497.6 | 1 | -427.3 | 0 | -416.9 | 0 | -406.7 | 0 | -395.0 | 0 |
| JG | 2 | **-530.3** | 5 | **-459.9** | 5 | **-447.8** | 7 | **-437.3** | 5 | **-424.1** | 7 |
| KP | 2 | -529.9 | 37 | -459.9 | 37 | -446.8 | 36 | -436.4 | 47 | -423.7 | 42 |
| MB | 5 | -493.0 | 0 | -425.0 | 1 | -409.8 | 0 | -402.5 | 0 | -393.3 | 0 |
| MM | 5 | -524.2 | 1 | -452.2 | 0 | -440.0 | 1 | -428.9 | 0 | -416.9 | 0 |
| JG | 5 | **-531.2** | 7 | **-461.5** | 8 | **-449.1** | 6 | **-438.2** | 8 | **-425.5** | 9 |
| KP | 5 | -530.3 | 24 | -459.8 | 37 | -446.8 | 31 | -436.9 | 33 | -422.4 | 33 |
| MB | 8 | -507.2 | 1 | -448.0 | 1 | -434.4 | 0 | -416.5 | 1 | -411.7 | 1 |
| MM | 8 | -528.5 | 0 | -460.2 | 1 | -448.5 | 1 | -435.9 | 1 | -423.5 | 1 |
| JG | 8 | **-531.7** | 23 | **-461.1** | 22 | **-449.2** | 25 | **-438.4** | 21 | **-425.2** | 23 |
| KP | 8 | -530.4 | 34 | -459.8 | 29 | -448.1 | 37 | -438.0 | 34 | -423.2 | 29 |
| MB | 10 | -520.3 | 2 | -452.7 | 2 | -438.7 | 2 | -427.4 | 2 | -414.6 | 3 |
| MM | 10 | -530.3 | 2 | -460.7 | 2 | -448.5 | 3 | -437.5 | 3 | -425.5 | 6 |
| JG | 10 | **-531.3** | 69 | **-461.4** | 65 | **-448.9** | 68 | **-438.4** | 76 | **-425.7** | 68 |
| KP | 10 | -529.7 | 36 | -460.0 | 33 | -448.4 | 36 | -437.2 | 28 | -424.4 | 32 |

Table 7: Table showing results on the 40x40 grid instance used in the UAI competitions. The line under the network reports the number of random variables, the maximum domain size of the variables, the number of functions and the treewidth. Alg: Algorithm, iB: i-bound, q10, q20, q50, q80 and q90: q values are chosen from the 10th, 20th, 50th, 80th and 90th percentile respectively, and Tm: time in seconds. The standard deviation for all methods was close to zero and therefore not reported.

| Alg | iB | Grids (1600,2,4800,113) | | | | | | | | | |
|---|---|---|---|---|---|---|---|---|---|---|---|
| | | q10 | Tm | q20 | Tm | q50 | Tm | q80 | Tm | q90 | Tm |
| MB | 2 | 136.0 | 1 | 1370.4 | 1 | 1587.4 | 0 | 1733.5 | 1 | 1949.3 | 1 |
| MM | 2 | -219.7 | 1 | 1040.8 | 1 | 1235.8 | 1 | 1404.5 | 1 | 1598.9 | 1 |
| JG | 2 | **-439.6** | 26 | 821.9 | 35 | **1017.8** | 30 | **1186.2** | 35 | **1381.4** | 28 |
| KP | 2 | -437.7 | 575 | **821.4** | 430 | 1020.4 | 406 | 1186.5 | 411 | 1383.3 | 332 |
| MB | 5 | -166.6 | 1 | 1078.4 | 2 | 1289.6 | 2 | 1439.3 | 2 | 1660.7 | 2 |
| MM | 5 | -403.9 | 2 | 857.3 | 2 | 1045.9 | 2 | 1222.0 | 2 | 1415.5 | 2 |
| JG | 5 | -441.4 | 51 | **819.9** | 51 | **1016.0** | 51 | 1184.3 | 45 | **1379.2** | 48 |
| KP | 5 | **-441.5** | 248 | 819.9 | 214 | 1016.3 | 164 | **1183.7** | 249 | 1379.2 | 237 |
| MB | 8 | -254.8 | 3 | 995.8 | 3 | 1178.7 | 3 | 1372.3 | 4 | 1557.7 | 2 |
| MM | 8 | -429.0 | 5 | 830.3 | 6 | 1024.8 | 6 | 1193.9 | 5 | 1389.6 | 5 |
| JG | 8 | -440.2 | 158 | 820.3 | 190 | **1016.0** | 160 | 1184.5 | 161 | 1380.6 | 196 |
| KP | 8 | **-441.3** | 251 | **819.6** | 202 | 1016.0 | 167 | **1183.9** | 222 | **1379.2** | 169 |
| MB | 10 | -318.1 | 8 | 951.7 | 8 | 1143.4 | 8 | 1308.7 | 6 | 1506.6 | 8 |
| MM | 10 | -437.0 | 10 | 824.1 | 12 | 1021.2 | 13 | 1186.1 | 11 | 1382.3 | 14 |
| JG | 10 | -438.0 | 453 | 823.9 | 396 | 1018.9 | 501 | 1184.8 | 393 | 1380.8 | 469 |
| KP | 10 | **-441.3** | 204 | **819.5** | 215 | **1016.0** | 170 | **1183.7** | 175 | **1379.1** | 216 |

Table 8: Table showing results on the Promedas OR Chain (for medical diagnosis) instance used in the UAI competitions. The line under the network reports the number of random variables, the maximum domain size of the variables, the number of functions and the treewidth. Alg: Algorithm, iB: i-bound, q10, q20, q50, q80 and q90: q values are chosen from the 10th, 20th, 50th, 80th and 90th percentile respectively, and Tm: time in seconds. The standard deviation for all methods was close to zero and therefore not reported.

| Alg | iB | Promedas (1953,2,1953,148) | | | | | | | | | |
|---|---|---|---|---|---|---|---|---|---|---|---|
| | | q10 | Tm | q20 | Tm | q50 | Tm | q80 | Tm | q90 | Tm |
| MB | 2 | -1968.0 | 1 | -1491.7 | 1 | -1431.5 | 1 | -1365.1 | 0 | -1274.5 | 1 |
| MM | 2 | -2456.8 | 1 | -1964.8 | 1 | -1881.8 | 1 | -1823.0 | 1 | -1738.2 | 1 |
| JG | 2 | -2953.6 | 51 | -2466.1 | 45 | -2362.5 | 56 | -2294.6 | 48 | -2218.3 | 51 |
| KP | 2 | **-3289.6** | 141 | **-2765.3** | 139 | **-2677.6** | 138 | **-2601.1** | 134 | **-2512.5** | 133 |
| MB | 5 | -3184.9 | 1 | -2661.6 | 1 | -2578.1 | 0 | -2493.0 | 1 | -2411.1 | 1 |
| MM | 5 | -3268.7 | 1 | -2748.8 | 1 | -2656.7 | 2 | -2584.8 | 1 | -2492.5 | 1 |
| JG | 5 | -3286.3 | 42 | -2764.6 | 37 | -2676.2 | 42 | -2599.3 | 44 | -2514.5 | 37 |
| KP | 5 | **-3291.9** | 124 | **-2770.8** | 116 | **-2681.2** | 150 | **-2605.3** | 95 | **-2519.0** | 168 |
| MB | 8 | -3198.4 | 1 | -2692.0 | 2 | -2599.4 | 1 | -2516.6 | 1 | -2427.1 | 2 |
| MM | 8 | -3284.6 | 3 | -2762.7 | 3 | -2670.1 | 4 | -2597.5 | 3 | -2509.3 | 3 |
| JG | 8 | -3280.3 | 91 | -2760.8 | 98 | -2670.7 | 126 | -2594.7 | 124 | -2509.8 | 112 |
| KP | 8 | **-3293.4** | 131 | **-2771.4** | 134 | **-2681.7** | 130 | **-2606.0** | 106 | **-2516.6** | 141 |
| MB | 10 | -3201.6 | 4 | -2688.4 | 4 | -2601.5 | 4 | -2528.6 | 5 | -2438.9 | 4 |
| MM | 10 | -3285.8 | 8 | -2763.1 | 8 | -2673.2 | 8 | -2601.2 | 8 | -2512.0 | 8 |
| JG | 10 | -3280.2 | 277 | -2756.4 | 322 | -2669.9 | 381 | -2592.3 | 395 | -2504.7 | 351 |
| KP | 10 | **-3293.1** | 93 | **-2770.8** | 115 | **-2680.3** | 101 | **-2606.8** | 119 | **-2519.3** | 131 |

Given an assignment $\boldsymbol{x}$, let $D$ be a decision variable which takes the value $d$ if $\sum_{g \in \mathcal{G}} g(\boldsymbol{x}) > 0$ and $\bar{d}$ otherwise. Given an assignment (test example) $\boldsymbol{x}$, our goal in rendering an adversarial attack is to modify it minimally such that the classifier decision flips, namely find an assignment $\boldsymbol{x}'$ such that the decision flips from $d$ to $\bar{d}$ and the distance between $\boldsymbol{x}$ and $\boldsymbol{x}'$ is minimized. Given an assignment $\boldsymbol{x}$, we can model the complement of the Hamming distance using a set of $n$ univariate functions $\mathcal{F} = \{f_1, \ldots, f_n\}$ where $f_i(x_i') = 1$ if $x_i' \in \boldsymbol{x}$ and 0 otherwise. Notice that if the hamming distance between $\boldsymbol{x}$ and $\boldsymbol{x}'$ is $k$ then $\sum_{i=1}^{n} f_i(x_i') = n - k$. Given an assignment $\boldsymbol{x}$ such that $\sum_{g \in \mathcal{G}} g(\boldsymbol{x}) > 0$

Table 9: Table showing results on the Pedigree 51 (for genetic linkage analysis) instance used in the UAI competitions. The line under the network reports the number of random variables, the maximum domain size of the variables, the number of functions and the treewidth. Alg: Algorithm, iB: i-bound, q10, q20, q50, q80 and q90: q values are chosen from the 10th, 20th, 50th, 80th and 90th percentile respectively, and Tm: time in seconds. The standard deviation for all methods was close to zero and therefore not reported.

| | | Pedigree (1152,5,1152,35) | | | | | | | | |
|---|---|---|---|---|---|---|---|---|---|---|
| Alg | iB | q10 | Tm | q20 | Tm | q50 | Tm | q80 | Tm | q90 | Tm |
| MB | 2 | -193.6 | 0 | -123.4 | 0 | -111.2 | 0 | -101.4 | 0 | -94.7 | 1 |
| MM | 2 | -310.2 | 1 | -235.6 | 1 | -222.0 | 1 | -213.4 | 1 | -204.9 | 0 |
| JG | 2 | -393.9 | 32 | -319.2 | 37 | -307.4 | 33 | -295.4 | 36 | -285.0 | 33 |
| KP | 2 | **-398.7** | 181 | **-323.6** | 214 | **-310.9** | 204 | **-299.6** | 166 | **-287.2** | 188 |
| MB | 5 | -342.0 | 0 | -268.9 | 1 | -260.2 | 1 | -245.0 | 0 | -237.0 | 0 |
| MM | 5 | -390.1 | 0 | -315.3 | 1 | -302.6 | 1 | -292.1 | 1 | -279.9 | 1 |
| JG | 5 | **-405.7** | 24 | **-330.8** | 32 | **-318.2** | 25 | **-307.3** | 28 | **-294.7** | 29 |
| KP | 5 | -401.2 | 196 | -324.8 | 139 | -312.6 | 134 | -301.0 | 150 | -290.8 | 133 |
| MB | 8 | -369.8 | 1 | -292.3 | 1 | -285.9 | 1 | -276.2 | 1 | -259.9 | 1 |
| MM | 8 | -401.3 | 3 | -327.6 | 3 | -313.9 | 3 | -302.0 | 2 | -288.4 | 3 |
| JG | 8 | **-404.8** | 79 | **-327.8** | 91 | **-317.1** | 84 | **-306.3** | 80 | **-292.3** | 81 |
| KP | 8 | -401.3 | 194 | -325.8 | 171 | -313.6 | 170 | -303.1 | 146 | -290.6 | 200 |
| MB | 10 | -373.8 | 4 | -300.9 | 3 | -284.9 | 4 | -279.4 | 4 | -269.2 | 4 |
| MM | 10 | -402.7 | 8 | -330.1 | 6 | **-316.4** | 8 | **-307.6** | 8 | **-294.4** | 8 |
| JG | 10 | **-404.0** | 298 | **-331.2** | 288 | -316.4 | 319 | -306.4 | 301 | -293.9 | 351 |
| KP | 10 | -401.4 | 140 | -324.9 | 211 | -313.8 | 193 | -304.0 | 183 | -292.5 | 225 |

Table 10: Table showing the average upper bound over 21000 test examples and the time required by the four algorithms MB, MM, JG and KP with i-bounds varying between 2 and 4 on the MNIST dataset using the Grid model for measuring distance and linear SVM as the classifier. Under the univariate distance model, the average upper bounds for MB and KP with i-bound set to 1 were $781.05\pm2.662$ and $\mathbf{780.8}\pm2.926$ respectively. Avg. UB: average upper bound, Std. UB: standard deviation of upper bound and Avg. Time: average time in seconds.

| | i-bound=2 | | | | i-bound=3 | | | | i-bound=4 | | | |
|---|---|---|---|---|---|---|---|---|---|---|---|---|
| | MB | MM | JG | KP | MB | MM | JG | KP | MB | MM | JG | KP |
| Avg. UB | 291.42 | 290.91 | **290.11** | 290.85 | 291.0 | 290.38 | **290.11** | 290.69 | 290.79 | 290.38 | **290.11** | 290.66 |
| Std. UB | 2.12 | 2.14 | 2.44 | 2.32 | 2.15 | 2.35 | 2.42 | 2.33 | 2.2 | 2.36 | 2.42 | 2.38 |
| Avg. Time | 0.0 | 0.0 | 2.8 | 0.6 | 0.0 | 0.0 | 3.0 | 0.4 | 0.0 | 0.0 | 3.3 | 0.3 |

and set of functions $\mathcal{F}$ and $\mathcal{G}$ as described above, we can now express the adversarial attack task as the following CMPE problem: $\max_{\boldsymbol{x}'} \sum_{f \in \mathcal{F}} f(\boldsymbol{x}')\ s.t.\ \sum_{g \in \mathcal{G}} g(\boldsymbol{x}') \leq 0$.

We can also use more sophisticated distance functions that take into account the *spatial arrangement* of pixels. For example, if we want pixels that are spatially close to take the same value, we can add the following set of pairwise functions to $\mathcal{F}$: $f_{i,j}(x_i, x_j) = 1.0$ if $X_i$ is a neighboring pixel of $X_j$ and $x_i = x_j$, and 0 otherwise. This will yield a graphical model whose primal graph is a grid.

We conducted experiments on the MNIST dataset [5] using the univariate and grid distance functions described above and linear support vector machine as the classifier. The dataset has 70,000 examples. We used 70% of the examples for training and 30% for testing. The accuracy of our classifier on the dataset was 97%. Note that a limitation of our approach in its current form is that we can only use multi-linear classifiers (aka graphical models without latent variables) and cannot use non-linear classifiers such as deep neural networks; however, we can use their linear or high treewidth approximations and this is a subject of future research.

Table 10 shows quantitative results comparing the quality of the upper bounds and time required for the univariate and grid distance functions. We see that on the univariate model, the KP based bounds yield exact answers (since the univariate problem is an instance of the knapsack problem) while the MPE based bounds are only slightly worse than the KP based bounds. On the grid models however, the MPE bounds are more accurate.

Figures 5, 6, 7, 8 and 9 show qualitative results. We consider two cases.

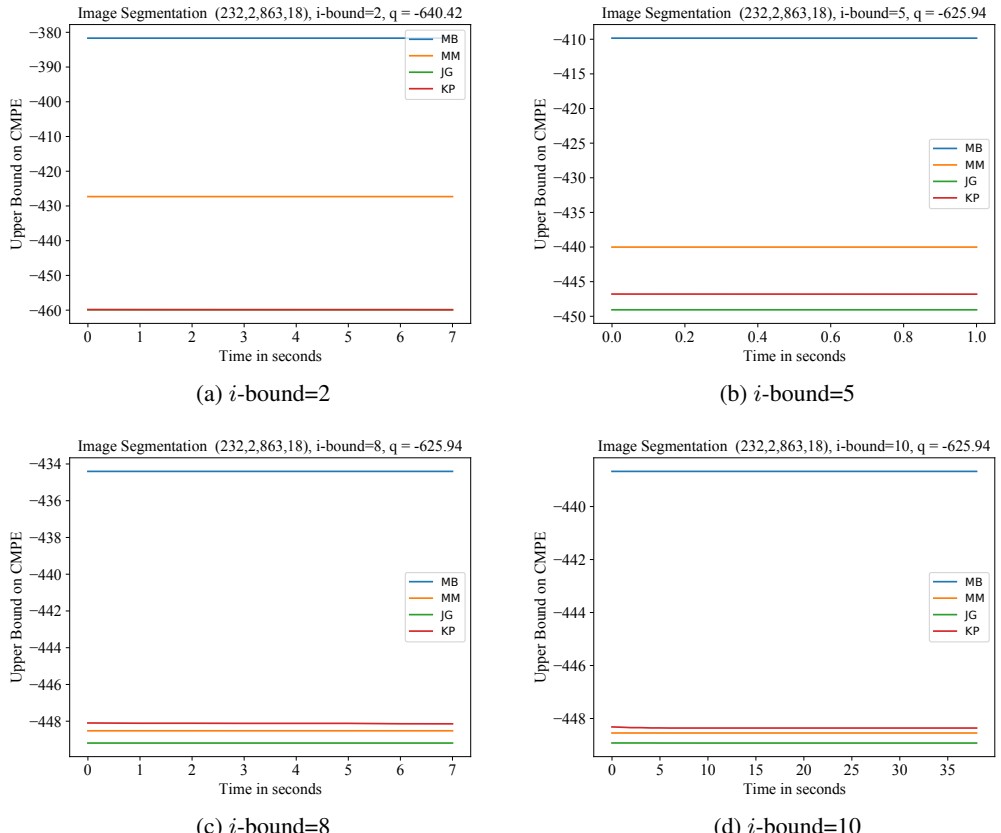

Figure 1: Figure showing upper bound (on CMPE) computed by the four schemes (MB, MM, JG and KP) as a function of time for different i-bounds and $q$ values on the Image Segmentation instance.

- **Case 1:** ($d$ to any) Given an image $I$ and a linear SVM $S$ such that $I$ is classified as digit $d$ by $S$, minimally change the pixels in $I$ to yield $I'$ such that the classification of $I'$ w.r.t. $S$ is no longer $d$ (namely it equals $\bar{d}$).

- **Case 2:** ($d$ to a specific $d'$) Given an image $I$ and a linear SVM $S$ such that $I$ is classified as digit $d$ by $S$, and another class $d'$ such that $d' \in \{0, \ldots, 9\}$ and $d' \neq d$, minimally change the pixels in $I$ to yield $I'$ such that $S$ classifies $I'$ as $d'$.

In Figure 5, we show qualitative results for Case 1. The first row shows the original images, the second row shows the modified images using the KP based algorithm for the univariate distance model and the third row shows the modified images using the MPE based algorithms for the Grid distance model. The qualitative results verify our intuition that the Grid model yields smoother deceptions than the univariate model.

In Figures 6, 7, 8 and 9, we show qualitative results for Case 2 where $d'$ equals 0, 4, 7 and 9 respectively. In each figure, the first row shows the original images, the second row shows the modified images using the KP based algorithm for the univariate distance model so that the classification changes to $d'$ and the third row shows the modified images using the MPE based algorithms for the Grid distance model such that the classification changes to $d'$.

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

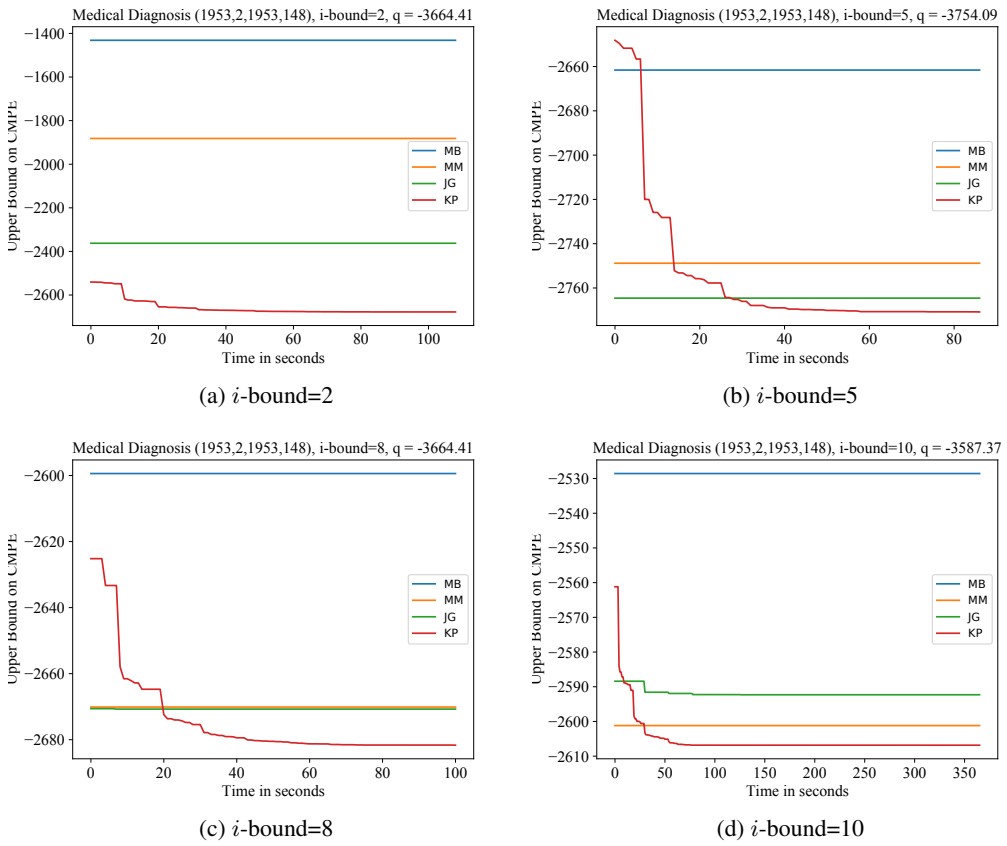

Figure 3: Figure showing upper bound (on CMPE) computed by the four schemes (MB, MM, JG and KP) as a function of time for different i-bounds and $q$ values on the Promedas OR Chain (for medical diagnosis) instance.

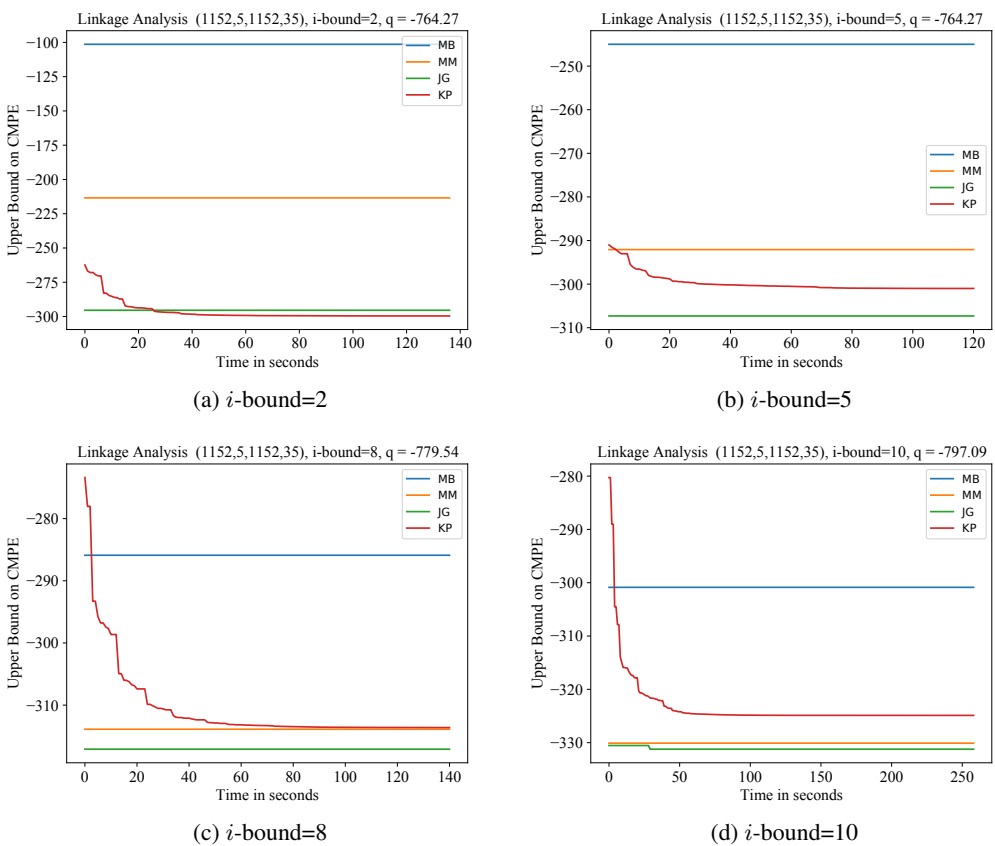

Figure 4: Figure showing upper bound (on CMPE) computed by the four schemes (MB, MM, JG and KP) as a function of time for different i-bounds and $q$ values on the Pedigree 51 (for genetic linkage analysis) instance.

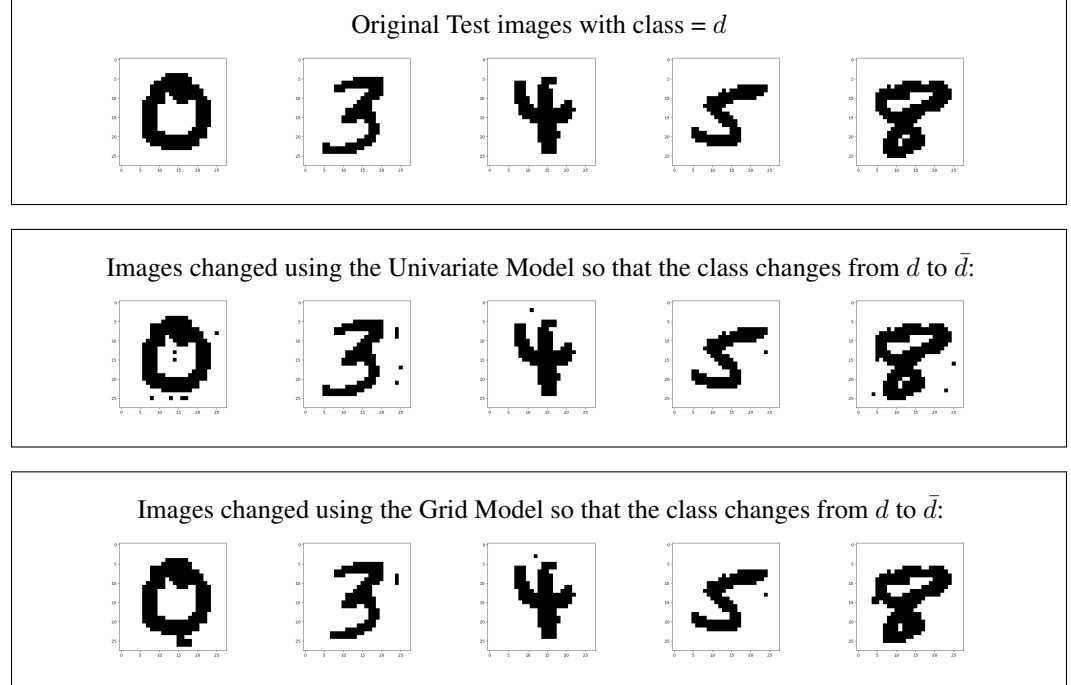

Figure 5: Qualitative results on MNIST dataset. Row 1 shows the original test images. Row 2 shows the images changed using the univariate model *so that the decision changes* from the original classification to any one of the other 9 digits. Row 3 shows the corresponding image changed using the Grid model *so that the decision changes* from the original classification to any one of the other 9 digits. For example, the zeros in the second and the third row will be classified as *not a zero* by the linear SVM. We notice that Grid model yields smoother decision flips (see the spurious black pixels in the images generated using the univariate model that are either absent or are rare in the Grid model).

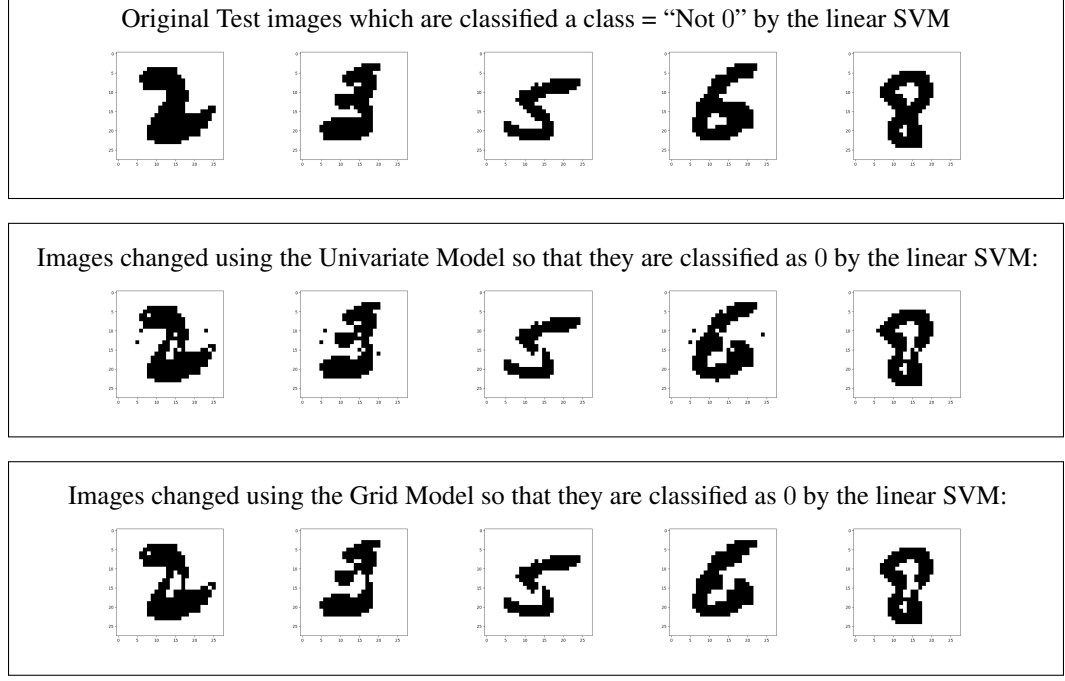

Figure 6: Qualitative results on MNIST dataset. Row 1 shows the original test images which are classified as "Not 0" by the linear SVM. Rows 2 and 3 show the images changed using the univariate and Grid model respectively *so that the decision made by the linear SVM changes* to 0. In other words, all images in the second and the third row will be classified as *zero* by the linear SVM. We notice that Grid model yields smoother decision flips; see the spurious black pixels in the images generated using the univariate model that are either absent or are rare in the Grid model.

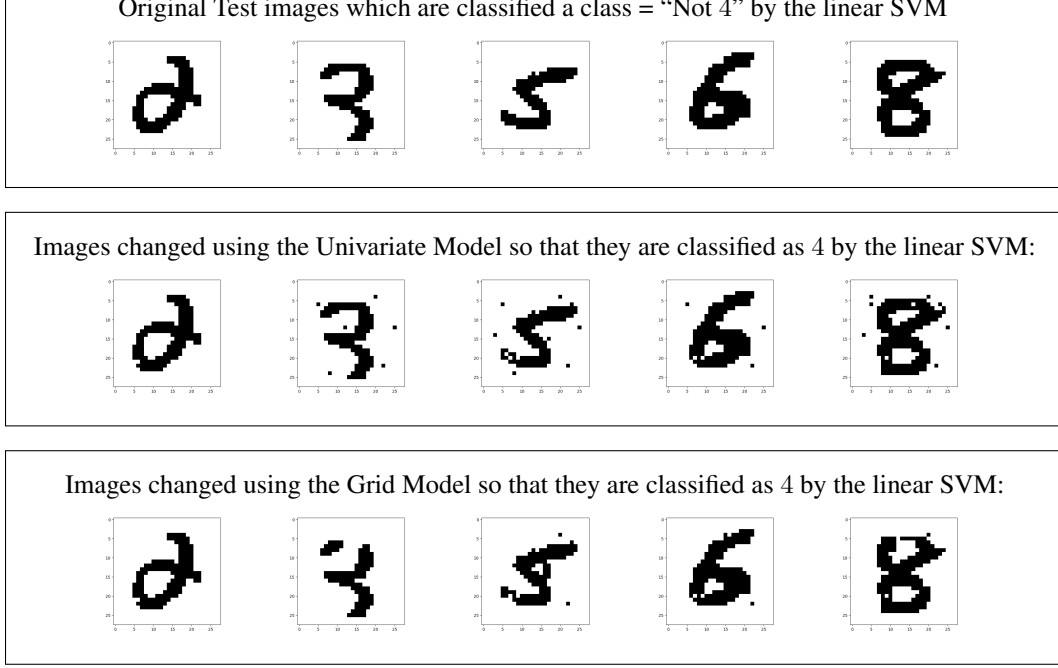

Figure 7: Qualitative results on MNIST dataset. Row 1 shows the original test images which are classified as "Not 4" by the linear SVM. Rows 2 and 3 show the images changed using the univariate and Grid model respectively *so that the decision made by the linear SVM changes* to 4. In other words, all images in the second and the third row will be classified as *4* by the linear SVM. We notice that Grid model yields smoother decision flips; see the spurious black pixels in the images generated using the univariate model that are either absent or are rare in the Grid model.

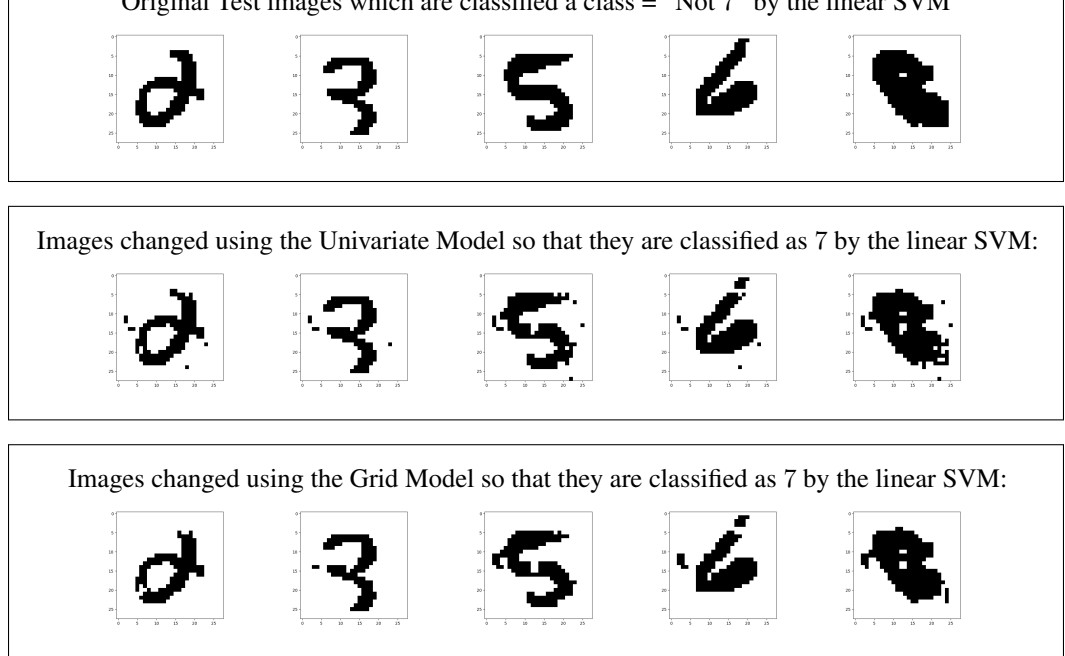

Figure 8: Qualitative results on MNIST dataset. Row 1 shows the original test images which are classified as "Not 7" by the linear SVM. Rows 2 and 3 show the images changed using the univariate and Grid model respectively *so that the decision made by the linear SVM changes* to 7. In other words, all images in the second and the third row will be classified as *7* by the linear SVM. We notice that Grid model yields smoother decision flips; see the spurious black pixels in the images generated using the univariate model that are either absent or are rare in the Grid model.

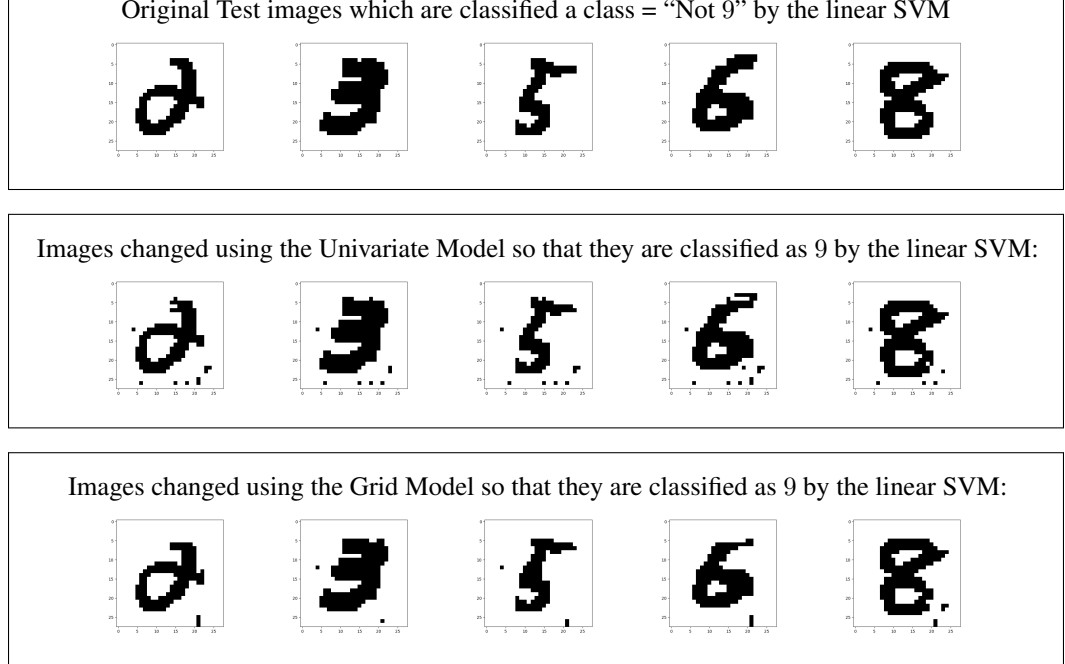

Figure 9: Qualitative results on MNIST dataset. Row 1 shows the original test images which are classified as "Not 9" by the linear SVM. Rows 2 and 3 show the images changed using the univariate and Grid model respectively *so that the decision made by the linear SVM changes* to 9. In other words, all images in the second and the third row will be classified as *9* by the linear SVM. We notice that Grid model yields smoother decision flips; see the spurious black pixels in the images generated using the univariate model that are either absent or are rare in the Grid model.