# OpenReview forum: "Novel Upper Bounds for the Constrained Most Probable Explanation Task"
_NeurIPS.cc/2021/Conference — NeurIPS 2021 Poster_

### Official Review · Reviewer_MHAL · 2021-07-15

**Rating:** 8
**Confidence:** 5

**Summary:**

The paper considers the constrained most probable explanation task in probabilistic graphical models and develops new upper bounding schemes for it. Unlike MPE, the CMPE task is NP-hard even for tree-shaped graphical models and therefore finding good upper bounds for it is definitely called for. Specifically, the idea is to use the Lagrangian relaxation to generate either a standard MPE problem or a knapsack problem. Experimental results of several CMPE benchmark domains demonstrate the effectiveness of the proposed upper bounding schemes.


**Ethical Concerns:**

There are no ethical concerns

**Ethics Review Area:**

["I don’t know"]

**Limitations And Societal Impact:**

The limitations of the proposed method are well addressed and there's no potential societal impact.

**Main Review:**

The paper is well written and organised. The quality of the presentation is overall very good. The two proposed upper bounding schemes are presented very clearly and examples provided help to get a better understanding. In the experimental section it may be worthwhile to include some plots of the computed upper bounds (as functions of the mini-bucket I-bounds for example).

Lagrangian relaxation is a well known technique that is widely used in operations research to deal efficiently with complicated constraints. Therefore, applying it to the CMPE task is quite natural. Although the extension is fairly straightforward it appears to be the first time it is done in this context. The experimental results demonstrate that the technique is quite effective in practice.

One thing the is perhaps missing is a discussion on the usefulness of these upper bounds. Although it is mentioned in the introduction that in principle they can be used to guide a heuristic search algorithm, it would be interesting to see how one would do it in practice (ie, precompile the heuristics in some way, generate them dynamically at each node in the search space, etc.).

Minor comment:
- In section 3.2 you also use a parameter "i" to bound the number of items in a partition. It's not exactly equivalent to the mini-bucket I-bound, so may be use a different letter for the parameter.


**Time Spent Reviewing:**

3 hours

---

> ### Author Response · Authors · 2021-08-10
> **Thank you for a thoughtful, detailed review**
>
> Thank you for a thoughtful, detailed review. Here are answers to your questions:
>
> ***
> Q1: One thing the is perhaps missing is a discussion on the usefulness of these upper bounds. Although it is mentioned in the introduction that in principle they can be used to guide a heuristic search algorithm, it would be interesting to see how one would do it in practice (ie, precompile the heuristics in some way, generate them dynamically at each node in the search space, etc.).
>
> Ans: You make an excellent point. We will include this in the discussion. As mentioned to other reviewers, our long term goal is to build a branch and bound solver and address relevant issues including (1) static versus dynamic upper bounds (run upper bounding at each node or periodically or just once); (2) variable ordering heuristics; (3) decimation schemes; etc.
>
> ***
> Q2:	 In section 3.2 you also use a parameter "i" to bound the number of items in a partition. It's not exactly equivalent to the mini-bucket I-bound, so may be use a different letter for the parameter.
>
> Again, thank you for your suggestion. We will use different letters so as not to confuse our bounds with mini-buckets.

---

### Official Review · Reviewer_av4V · 2021-07-16

**Rating:** 7
**Confidence:** 4

**Summary:**

This paper presents two novel schemes for upper bounding the CMPE problem. The first relaxes the global constraint via Lagrangian relaxation and the second reduces the CMPE to an MCKP problem.

**Limitations And Societal Impact:**

See my comments in "Main review"

**Main Review:**

The paper is well-written and contributions are solid. I personally like the paper but I have several minor concerns:

(1) How tight are the two new bounds? It is unclear to me how much these two new bounds are better than the naive upper bound obtained by removing the constraint in Eq (`1).

(2) The paper claims the usages of the new upper bounds are  (1) generating heuristic
near-optimal solutions; and (2) pruning the search space of branch and bound methods and thus improving their efficiency. However, there are no experiments provided about these two usage cases in the paper. Therefore, I am not sure how much these two new bounds can help in practices.

(3) The time and space complexities of the second scheme seem rather high, it looks to me that it will significantly hinder the use of such scheme in the usage of pruning search space in a complete search engine like branch and bound.

**Time Spent Reviewing:**

1

---

> ### Author Response · Authors · 2021-08-10
> **Thank you for asking important questions and a thoughtful review.**
>
> Thank you for asking important questions and a thoughtful review.
> Here are the answers to your questions.
> ***
> Q1: How tight are the two new bounds? It is unclear to me how much these two new bounds are better than the naive upper bound obtained by removing the constraint in Eq (`1).
>
> * Ans: The bounds given in the paper are provably much tighter than the bound obtained by removing the constraint; all you have to do is set lambda to zero in Equation 3 and you get the naive bound you describe. Compare the q80 numbers with the q20 numbers (see the tables in the paper) to get a sense of how loose the naive upper bound can be in practice. We will include the MPE values for each problem in the respective tables; this is easy to do. Thank you for the suggestion.
>
>
> ***
> Q2:  The paper claims the usages of the new upper bounds are (1) generating heuristic near-optimal solutions; and (2) pruning the search space of branch and bound methods and thus improving their efficiency. However, there are no experiments provided about these two usage cases in the paper. Therefore, I am not sure how much these two new bounds can help in practices.
>
> * Ans: For (1), namely generating heuristic near-optimal solutions, we have provided qualitative results on the mnist dataset in the supplement. For (2), namely pruning search space of branch and bound is part of our future work. Our long term goal is to develop a specialized branch and bound algorithm for the CMPE task that has better performance than off-the-shelf integer linear programming solvers.
>
> ***
> Q3: The time and space complexities of the second scheme seem rather high, it looks to me that it will significantly hinder the use of such scheme in the usage of pruning search space in a complete search engine like branch and bound.
>
> * Ans: As mentioned in the paper  (see page 6), the time and space complexity of the second scheme can be controlled by a parameter “i”. The complexity of the scheme is exponential in “i” but since “i” is a constant, the method runs in polynomial time when MCKP is tractable. When MCKP is not tractable for a given “i”, we can approximate it using fast upper bounding methods for MCKP (see the book on knapsack problems). To reiterate, all upper bounding schemes presented in the paper require polynomial time.

---

### Official Review · Reviewer_9jTV · 2021-07-17

**Rating:** 7
**Confidence:** 3

**Summary:**

The authors present two novel approaches for upper bounding the constrained most probable explanation (CMPE) task in graphical models.  The CMPE is a variation of the MPE or MAP estimation problem in discrete graphical models, where one must find the most likely configuration of the graphical model subject to a constraint on the chosen configuration.  The authors make use of lagrangian relaxation to introduce two new methods.  One works by relaxing the CMPE to an MPE, which can be solved exactly or approximately, using the solution to compute the upper bound, tightening the bound via subgradient descent and repeating.  The other works in a similar fashion by relaxing the CMPE to a multi choice knapsack problem.

As far as I can tell, these contributions are both novel and significant.

**Limitations And Societal Impact:**

This doesn't seem relevant.

**Main Review:**

Overall, I thought this was a very well written paper.  The problem was tightly described, the contributions (and limitations) were clearly presented, the technical results were clear and concise and the results conveyed the efficacy and limitations of the proposed approaches. And to the best of my knowledge, these appear to be the first generally tractable methods to upper bound the CMPE problem.

A few comments/questions:
-Is the assumption that the log linear models F (MPE) and G (constraint) have the same primal graph limiting? And does it obscure the real trade-off between the two methods?  If the graphs weren't the same, would you choose the MCKP approximation when the model G has higher treewidth and the MPE-based approximation when F has higher treewidth?
-It was not obvious to me that \sum_{g \in G} g(x) <= q in the MPE based approximation when the MPE model is not tractable and must be approximated. Can you provide some additional color on this claim.
-Did you run into any convergence issues in the case where the MPE approximation was not tractable?
-What is the relationship between i and k on lines 162 and 163, or is that a typo?
-Line 227 I don't think you mean "duplicating each variable two times".


**Time Spent Reviewing:**

1

---

> ### Author Response · Authors · 2021-08-10
> **Thank you for reading the paper carefully, asking important questions and finding typos.**
>
> Thank you for reading the paper carefully, asking important questions and finding typos.
>
> Here are the answers to your questions:
> ***
> Q1: Is the assumption that the log linear models F (MPE) and G (constraint) have the same primal graph limiting? And does it obscure the real trade-off between the two methods? If the graphs weren't the same, would you choose the MCKP approximation when the model G has higher treewidth and the MPE-based approximation when F has higher treewidth?
>
> Ans: The answer to all these questions is ''No.'' The assumption is not limiting and is made without loss of generality. Both methods (MPE and MCKP) mix the primal graphs of the constraint and the objective (in other words, they operate on the union of the two primal graphs). Thus, it does not matter if F has higher treewidth or G has higher treewidth; what matters is the treewidth of the combined primal graph.
> ***
>
> Q2: It was not obvious to me that \sum_{g \in G} g(x) <= q in the MPE based approximation when the MPE model is not tractable and must be approximated. Can you provide some additional color on this claim.
>
> Ans: \sum_{g \in G} g(x) <= q is the global constraint for the CMPE task. We obtain the Lagrangian relaxation of the CMPE task by relaxing this global constraint using a Lagrange multiplier. Given a value for this multiplier, this relaxation converts the original CMPE task to the MPE task which we can solve either exactly or upper bound it using MB/MM/JG depending on the treewidth of the combined primal graph.
>
> ***
>
> Q3. Did you run into any convergence issues in the case where the MPE approximation was not tractable?
>
> Ans: In our experiments (as mentioned in the paper and supplement), we ran the iterative algorithms (MPE approximators and MCKP bounding methods) for a fixed number of iterations (40/100) or until convergence. We did not run into convergence issues. See the plots in the supplement which show fast convergence.
> ***
> Q4: What is the relationship between i and k on lines 162 and 163, or is that a typo?
>
> Ans: Thank you for pointing this out. This is a typo and we will fix it.
> ***
> Q5: Line 227 I don't think you mean "duplicating each variable two times".
>
> Ans: Thanks again. We will correct this.

---

### Official Review · Reviewer_NaDK · 2021-07-25

**Rating:** 7
**Confidence:** 4

**Summary:**

This paper proposes tractable strategies for upper bounding the value of constrained MPE problems. Two strategies are developed based on Lagrangian relaxation. One is to completely eliminate the constraint, converting the problem to a straight MPE problem. The second one involves the strategic application of dual decomposition to convert the problem instance to one of a multiple choice knapsack problem. The approaches are evaluated on datasets from the UAI inference challenge as well as adversarial input modification tasks.

**Limitations And Societal Impact:**

No concerns.

**Main Review:**

This is a very well written paper that addresses an reasonably important tasks through non-trivial means. The ideas are clearly communicated and easy to follow. Regarding novelty --- I found the MCKP bound to be rather clever and this thankfully translates to interesting computational results as well.

I have a few points of criticism.
First, the final paragraph on page 4 (lines 170 - 174) imply somehow that LP-based bounds are unworthy of consideration because of the weak polynomial complexity of linear programming. I find this point extremely contrived given that the first bound simply assumes that the unconstrained MPE problem will be tractable or otherwise the recourse are approximation algorithms with running time proportional to 2^i  If one is comfortable accepting MPE tractability, accepting LP tractability should be a much easier pill to swallow. This is actually not a minor point, since the domination of the LP bound by the first Lagrangian bound holds only if the MPE is solved exactly. Otherwise, no straightforward claims can be made. The LP bound should not only not be discarded so lightly but is probably the most natural baseline for the evaluations of this paper.

Second, one thing that is actually confusing on first read is the way that the way that the algorithms under evaluation are introduced on page 7, section 4. At first I thought that MG/MM/JG are existing bounding approaches, since they were discussed in the prior art section  (which might leave a confusing impression as the reader might have forgotten exactly what the situation is at that point), then I wondered where are the evaluations of the first bound, so it took some back and forth until things clicked. I think it's worth calling this out explicitly in a single sentence that MG/MM/JG are evaluations of the bound in eq. 3.

Finally, I think it's worth considering to move all the tabular data to the appendix and instead having plots to illustrate the specific points that are made in section 4. I think going through all these tables multiple times makes for a rather tedious reading experience, yet there are rather interesting observations here that shouldn't be missed, so having a more condensed representation might make sense.

Overall, I would say this is a solid paper. It can be further strengthened by the addition of an LP baseline. The other issues are cosmetic.



**Time Spent Reviewing:**

999

---

> ### Author Response · Authors · 2021-08-10
> **Thank you for your on-point criticisms and suggestions to improve the paper**
>
> Thank you for your comments and suggestions. We appreciate your on-point criticisms and will address them in the camera ready version.
>
> * We will address your criticism about LP-bound by including LP results (time/quality of bounds). We were not discarding LP bounds but trying to come up with faster alternatives; LP can be very slow when the number of variables is large. Specifically, the goal of the paper is to investigate (upper bounding) algorithms that have strong polynomial time guarantees and are as close as possible and in some cases better than LP bounds. Informally, Theorem 1 says that: (1) When MPE is tractable, we will typically beat LP both time wise and quality wise; and (2) When MPE is intractable, we will typically beat LP time wise by using algorithms such as MB/MM/JG but may not beat LP quality wise. But again, we want to stress the goal: strong poly-time guarantees with a long term goal of building specialized branch and bound solvers for CMPE and related problems. It is well known in combinatorial optimization/operations research literature that the computational complexity of LP can be quite high when global constraints such as the ones in CMPE are present and as a result using specialized upper bound solvers that yield faster solutions (even if they are inferior to LP) is a good idea.
>
> * Thank you for the suggestion about explicitly stating that MB/MM/JG are evaluations of bounds in Eq. (3). We have included a para about it in section 3 but will remind the reader in the experiments section.
>
> * Thank you for the suggestion about experimental results. We would appreciate you elaborating on this aspect. We have an extra page if the paper is accepted and we will use it for improving the experiments section.

---

### Decision · Program_Chairs · 2021-09-27

**Decision:**

Accept (Poster)

**Comment:**

The authors present two approaches for bounding "constrained" most probable explanation (CMPE) tasks in graphical models, in which we seek an MPE solution to one model, constrained to a subset of configurations by another model.  Both methods are simple (in a good way), relaxing the CMPE to an unconstrained MPE problem, or to a multi-choice knapsack problem, then using Lagrangian optimization to tighten the resulting bounds.  Reviewers were unanimously positive, highlighting the novelty of the work as a strength.  Several reviewers did bring up points that should be addressed in a final version, however, including comparison to LP-based techniques, and some issues with the presentation (see individual reviews for details).